# Diffusion Models Meet Contextual Bandits with Large Action Spaces

## Abstract

Efficient exploration in contextual bandits is crucial due to their large action space, where uninformed exploration can lead to computational and statistical inefficiencies. However, the rewards of actions are often correlated, which can be leveraged for more efficient exploration. In this work, we use pre-trained diffusion model priors to capture these correlations and develop diffusion Thompson sampling (dTS). We establish both theoretical and algorithmic foundations for dTS. Specifically, we derive efficient posterior approximations (required by dTS) under a diffusion model prior, which are of independent interest beyond bandits and reinforcement learning. We analyze dTS in linear instances and provide a Bayes regret bound highlighting the benefits of using diffusion models as priors. Our experiments validate our theory and demonstrate dTS's favorable performance.

## 1   Introduction

A *contextual bandit* is a popular and practical framework for online learning under uncertainty [Li et al., 2010]. In each round, an agent observes a *context*, takes an *action*, and receives a *reward* based on the context and action. The goal is to maximize the expected cumulative reward over $n$ rounds, striking a balance between exploiting actions with high estimated rewards from available data and exploring other actions to improve current estimates. This trade-off is often addressed using either *upper confidence bound (UCB)* [Auer et al., 2002] or *Thompson sampling (TS)* [Scott, 2010].

The action space in contextual bandits is often large, resulting in less-than-optimal performance with standard exploration strategies. Luckily, actions usually exhibit correlations, making efficient exploration possible as one action may inform the agent about other actions. In particular, Thompson sampling offers remarkable flexibility, allowing its integration with informative priors [Hong et al., 2022b] that capture these correlations. Inspired by the achievements of diffusion models [Sohl-Dickstein et al., 2015, Ho et al., 2020], which effectively approximate complex distributions [Dhariwal and Nichol, 2021, Rombach et al., 2022], this work captures action correlations by employing diffusion models as priors in contextual Thompson sampling.

We illustrate the idea using video streaming. The objective is to optimize watch time for a user $j$ by selecting a video $i$ from a catalog of $K$ videos. Users $j$ and videos $i$ are associated with context vectors $x_j$ and unknown video parameters $\theta_i$, respectively. User $j$'s expected watch time for video $i$ is linear as $x_j^\top \theta_i$. Then, a natural strategy is to independently learn video parameters $\theta_i$ using LinTS or LinUCB [Agrawal and Goyal, 2013a, Abbasi-Yadkori et al., 2011], but this proves statistically inefficient for larger $K$. Fortunately, the reward when recommending a movie can provide informative insights into other movies. To capture this, we leverage offline estimates of video parameters denoted by $\hat{\theta}_i$ and build a diffusion model on them. This diffusion model approximates the video parameter distribution, capturing their dependencies. This model enriches contextual Thompson sampling as a prior, effectively capturing complex video dependencies while ensuring computational efficiency.

Submitted to 38th Conference on Neural Information Processing Systems (NeurIPS 2024). Do not distribute.

We introduce a framework for contextual bandits with diffusion model priors, upon which we develop diffusion Thompson sampling (dTS) that is both computationally and statistically efficient. dTS requires *fast updates of the posterior* and *fast sampling from the posterior*, both of which are achieved through our novel efficient posterior approximations. These approximations become exact when both the diffusion model and likelihood are linear. We establish a bound on dTS's Bayes regret for this specific case, highlighting the advantages of using diffusion models as priors. Our empirical evaluations validate our theory and demonstrate dTS's strong performance across various settings.

Diffusion models were applied in offline decision-making [Ajay et al., 2022, Janner et al., 2022, Wang et al., 2022], but their use in online learning was only recently explored by Hsieh et al. [2023], who focused on *multi-armed bandits without theoretical guarantees*. Our work extends Hsieh et al. [2023] in two ways. First, we apply the concept to the broader contextual bandit, which is more practical and realistic. Second, we demonstrate that with diffusion models parametrized by linear score functions and linear rewards, we can derive exact closed-form posteriors without approximations. These exact posteriors are valuable as they enable theoretical analysis (unlike Hsieh et al. [2023], who did not provide theoretical guarantees) and motivate efficient approximations for non-linear score functions in contextual bandits, addressing gaps in Hsieh et al. [2023]'s focus on multi-armed bandits.

A key contribution, beyond applying diffusion models in contextual bandits, is the efficient *computation* and *sampling* of the posterior distribution of a $d$-dimensional parameter $\theta \mid H_t$, with $H_t$ representing the data, when using a diffusion model prior on $\theta$. This is relevant not only to bandits and reinforcement learning but also to a broader range of applications [Chung et al., 2022]. To motivate our approximations, we start with exact closed-form solutions for cases where both the score functions of the diffusion model and the likelihood are linear. These solutions form the basis for our approximations for non-linear score functions, demonstrating both strong empirical performance and computational efficiency. Our approach avoids the computational burden of heavy approximate sampling algorithms required for each latent parameter. For a detailed comparison with existing studies, see Appendix A, where we discuss diffusion models in decision-making, structured bandits, approximate posteriors, and more.

## 2 Setting

The agent interacts with a *contextual bandit* over $n$ rounds. In round $t \in [n]$, the agent observes a *context* $X_t \in \mathcal{X}$, where $\mathcal{X} \subseteq \mathbb{R}^d$ is a *context space*, it takes an *action* $A_t \in [K]$, and then receives a stochastic reward $Y_t \in \mathbb{R}$ that depends on both the context $X_t$ and the taken action $A_t$. Each action $i \in [K]$ is associated with an *unknown action parameter* $\theta_{*,i} \in \mathbb{R}^d$, so that the reward received in round $t$ is $Y_t \sim P(\cdot \mid X_t; \theta_{*,A_t})$, where $P(\cdot \mid x; \theta_{*,i})$ is the reward distribution of action $i$ in context $x$. Throughout the paper, we assume that the reward distribution is parametrized as a generalized linear model (GLM) [McCullagh and Nelder, 1989]. That is, for any $x \in \mathcal{X}$, $P(\cdot \mid x; \theta_{*,i})$ is an exponential-family distribution with mean $g(x^\top \theta_{*,i})$, where $g$ is the mean function. For example, we recover linear bandits when $P(\cdot \mid x; \theta_{*,i}) = \mathcal{N}(\cdot; x^\top \theta_{*,i}, \sigma^2)$ where $\sigma > 0$ is the observation noise. Similarly, we recover logistic bandits [Filippi et al., 2010] if we let $g(u) = (1 + \exp(-u))^{-1}$ and $P(\cdot \mid x; \theta_{*,i}) = \mathrm{Ber}(g(x^\top \theta_{*,i}))$, where $\mathrm{Ber}(p)$ be the Bernoulli distribution with mean $p$.

We consider the *Bayesian* bandit setting [Russo and Van Roy, 2014, Hong et al., 2022b], where the action parameters $\theta_{*,i}$ are assumed to be sampled from a *known* prior distribution. We proceed to define this prior distribution using a diffusion model. The correlations between the action parameters $\theta_{*,i}$ are captured through a diffusion model, where they share a set of $L$ consecutive *unknown latent parameters* $\psi_{*,\ell} \in \mathbb{R}^d$ for $\ell \in [L]$. Precisely, the action parameter $\theta_{*,i}$ depends on the $L$-th latent parameter $\psi_{*,L}$ as $\theta_{*,i} \mid \psi_{*,1} \sim \mathcal{N}(f_1(\psi_{*,1}), \Sigma_1)$, where the *score function* $f_1 : \mathbb{R}^d \to \mathbb{R}^d$ is *known*. Also, the $\ell-1$-th latent parameter $\psi_{*,\ell-1}$ depends on the $\ell$-th latent parameter $\psi_{*,\ell}$ as $\psi_{*,\ell-1} \mid \psi_{*,\ell} \sim \mathcal{N}(f_\ell(\psi_{*,\ell}), \Sigma_\ell)$, where the score function $f_\ell : \mathbb{R}^d \to \mathbb{R}^d$ is known. Finally, the $L$-th latent parameter $\psi_{*,L}$ is sampled as $\psi_{*,L} \sim \mathcal{N}(0, \Sigma_{L+1})$. We summarize this model in (1) and its graph in Fig. 1.

$$
\begin{aligned}
\psi_{*,L} &\sim \mathcal{N}(0, \Sigma_{L+1}), & (1)\\
\psi_{*,\ell-1} \mid \psi_{*,\ell} &\sim \mathcal{N}(f_\ell(\psi_{*,\ell}), \Sigma_\ell), & \forall \ell \in [L]/\{1\},\\
\theta_{*,i} \mid \psi_{*,1} &\sim \mathcal{N}(f_1(\psi_{*,1}), \Sigma_1), & \forall i \in [K],\\
Y_t \mid X_t, \theta_{*,A_t} &\sim P(\cdot \mid X_t; \theta_{*,A_t}), & \forall t \in [n].
\end{aligned}
$$

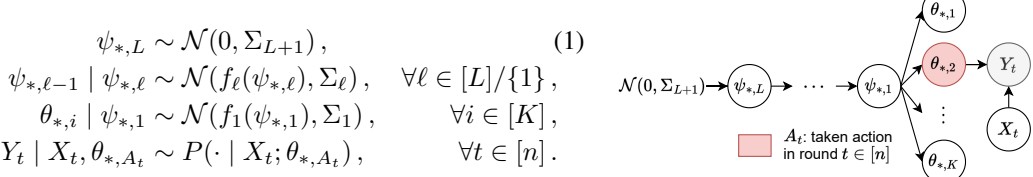

Figure 1: Graphical model of (1).

The model in (1) represents a Bayesian bandit, where the agent interacts with a bandit instance defined by $\theta_{*,i}$ over $n$ rounds (4-th line in (1)). These action parameters $\theta_{*,i}$ are drawn from the generative process in the first 3 lines of (1). In practice, (1) can be built by pre-training a diffusion model on offline estimates of the action parameters $\theta_{*,i}$ [Hsieh et al., 2023].

A natural goal for the agent in this Bayesian framework is to minimize its *Bayes regret* [Russo and Van Roy, 2014] that measures the expected performance across multiple bandit instances $\theta_* = (\theta_{*,i})_{i \in [K]}$,

$$\mathcal{BR}(n) = \mathbb{E}\Big[ \sum_{t=1}^n r(X_t, A_{t,*}; \theta_*) - r(X_t, A_t; \theta_*) \Big], \tag{2}$$

where the expectation in (2) is taken over all random variables in (1). Here $r(x, i; \theta_*) = \mathbb{E}_{Y \sim P(\cdot | x; \theta_{*,i})}[Y]$ is the expected reward of action $i$ in context $x$ and $A_{t,*} = \arg\max_{i \in [K]} r(X_t, i; \theta_*)$ is the optimal action in round $t$. The Bayes regret is known to capture the benefits of using informative priors, and hence it is suitable for our problem.

## 3 Diffusion contextual Thompson sampling

We design Thompson sampling that samples the latent and action parameters hierarchically [Lindley and Smith, 1972]. Precisely, let $H_t = (X_k, A_k, Y_k)_{k \in [t-1]}$ be the history of all interactions up to round $t$ and let $H_{t,i} = (X_k, A_k, Y_k)_{\{k \in [t-1]; A_k = i\}}$ be the history of interactions *with action $i$* up to round $t$. To motivate our algorithm, we decompose the posterior $\mathbb{P}(\theta_{*,i} = \theta \,|\, H_t)$ recursively as

$$\mathbb{P}(\theta_{*,i} = \theta \,|\, H_t) = \int_{\psi_{1:L}} Q_{t,L}(\psi_L) \prod_{\ell=2}^L Q_{t,\ell-1}(\psi_{\ell-1} \,|\, \psi_\ell) P_{t,i}(\theta \,|\, \psi_1) \, d\psi_{1:L}, \quad \text{where} \tag{3}$$

$Q_{t,L}(\psi_L) = \mathbb{P}(\psi_{*,L} = \psi_L \,|\, H_t)$ is the *latent-posterior* density of $\psi_{*,L} \,|\, H_t$. Moreover, for any $\ell \in [2 : L]$, $Q_{t,\ell-1}(\psi_{\ell-1} \,|\, \psi_\ell) = \mathbb{P}(\psi_{*,\ell-1} = \psi_{\ell-1} \,|\, H_t, \psi_{*,\ell} = \psi_\ell)$ is the *conditional latent-posterior* density of $\psi_{*,\ell-1} \,|\, H_t, \psi_{*,\ell} = \psi_\ell$. Finally, for any action $i \in [K]$, $P_{t,i}(\theta \,|\, \psi_1) = \mathbb{P}(\theta_{*,i} = \theta \,|\, H_{t,i}, \psi_{*,1} = \psi_1)$ is the *conditional action-posterior* density of $\theta_{*,i} \,|\, H_{t,i}, \psi_{*,1} = \psi_1$.

The decomposition in (3) inspires hierarchical sampling. In round $t$, we initially sample the $L$-th latent parameter as $\psi_{t,L} \sim Q_{t,L}(\cdot)$. Then, for $\ell \in [L]/\{1\}$, we sample the $\ell - 1$-th latent parameter given that $\psi_{*,\ell} = \psi_{t,\ell}$, as $\psi_{t,\ell-1} \sim Q_{t,\ell-1}(\cdot \,|\, \psi_{t,\ell})$. Lastly, given that $\psi_{*,1} = \psi_{t,1}$, each action parameter is sampled *individually* as $\theta_{t,i} \sim P_{t,i}(\theta \,|\, \psi_{t,1})$. This is possible because action parameters $\theta_{*,i}$ are conditionally independent given $\psi_{*,1}$. This leads to Algorithm 1, named **d**iffusion **T**hompson **S**ampling (dTS). dTS requires sampling from the $K + L$ posteriors $P_{t,i}$ and $Q_{t,\ell}$. Thus we start by providing an efficient recursive scheme to express these posteriors using known quantities. We note that these expressions do not necessarily lead to closed-form posteriors and approximation might be needed. First, the conditional action-posterior $P_{t,i}(\cdot \,|\, \psi_1)$ can be written as

$$P_{t,i}(\theta \,|\, \psi_1) \propto \prod_{k \in S_{t,i}} P(Y_k \,|\, X_k; \theta) \mathcal{N}(\theta; f_1(\psi_1), \Sigma_1), \tag{4}$$

where $S_{t,i} = \{\ell \in [t-1], A_\ell = i\}$ are the rounds where the agent takes action $i$ up to round $t$. Moreover, let $\mathcal{L}_\ell(\psi_\ell) = \mathbb{P}(H_t \,|\, \psi_{*,\ell} = \psi_\ell)$ be the likelihood of observations up to round $t$ given that $\psi_{*,\ell} = \psi_\ell$. Then, for any $\ell \in [L]/\{1\}$, the $\ell - 1$-th conditional latent-posterior $Q_{t,\ell-1}(\cdot \,|\, \psi_\ell)$ is

$$Q_{t,\ell-1}(\psi_{\ell-1} \,|\, \psi_\ell) \propto \mathcal{L}_{\ell-1}(\psi_{\ell-1}) \mathcal{N}(\psi_{\ell-1}, f_\ell(\psi_\ell), \Sigma_\ell), \tag{5}$$

and $Q_{t,L}(\psi_L) \propto \mathcal{L}_L(\psi_L) \mathcal{N}(\psi_L, 0, \Sigma_{L+1})$. All the terms above are known, except the likelihoods $\mathcal{L}_\ell(\psi_\ell)$ for $\ell \in [L]$. These are computed recursively as follows. First, the basis of the recursion is

$$\mathcal{L}_1(\psi_1) = \prod_{i=1}^K \int_{\theta_i} \prod_{k \in S_{t,i}} P(Y_k \,|\, X_k; \theta_i) \mathcal{N}(\theta_i; f_1(\psi_1), \Sigma_1) \, d\theta_i. \tag{6}$$

Then for $\ell \in [L]/\{1\}$, the recursive step is $\mathcal{L}_\ell(\psi_\ell) = \int_{\psi_{\ell-1}} \mathcal{L}_{\ell-1}(\psi_{\ell-1}) \mathcal{N}(\psi_{\ell-1}; f_\ell(\psi_\ell), \Sigma_\ell) \, d\psi_{\ell-1}$.

All posterior expressions above use known quantities $(f_\ell, \Sigma_\ell, P(y \,|\, x; \theta))$. However, these expressions typically need to be approximated, except when the score functions $f_\ell$ are linear and the reward distribution $P(\cdot \,|\, x; \theta)$ is linear-Gaussian, where closed-form solutions can be obtained with careful derivations. These approximations are not trivial, and prior studies often rely on computationally intensive approximate sampling algorithms. In the following sections, we explain how we derive our efficient approximations which are motivated by the closed-form solutions of linear instances.

**Algorithm 1** dTS: **d**iffusion **T**hompson **S**ampling

**Input:** Prior: $f_\ell, \ell \in [L]$, $\Sigma_\ell, \ell \in [L+1]$, and $P$.
**for** $t = 1, \ldots, n$ **do**

  Sample $\psi_{t,L} \sim Q_{t,L}$ (requires fast approximate posterior update and sampling)
  **for** $\ell = L, \ldots, 2$ **do**
    Sample $\psi_{t,\ell-1} \sim Q_{t,\ell-1}(\cdot \mid \psi_{t,\ell})$ (requires fast approximate posterior update and sampling)
  **for** $i = 1, \ldots, K$ **do**
    Sample $\theta_{t,i} \sim P_{t,i}(\cdot \mid \psi_{t,1})$ (requires fast approximate posterior update and sampling)
  Take action $A_t = \mathrm{argmax}_{i \in [K]} r(X_t, i; \theta_t)$, where $\theta_t = (\theta_{t,i})_{i \in [K]}$
  Receive reward $Y_t \sim P(\cdot \mid X_t; \theta_{*,A_t})$ and update posteriors $Q_{t+1,\ell}$ and $P_{t+1,i}$.

## 3.1 Linear diffusion model

Assume the score functions $f_\ell$ are linear such as $f_\ell(\psi_{*,\ell}) = W_\ell \psi_{*,\ell}$ for $\ell \in [L]$, where $W_\ell \in \mathbb{R}^{d \times d}$ are *known mixing matrices*. Then, (1) becomes a linear Gaussian system (LGS) [Bishop, 2006] in this case. This model is important, both in theory and practice. For theory, it leads to closed-form posteriors when the reward distribution is linear-Gaussian as $P(\cdot \mid x; \theta_{*,i}) = \mathcal{N}(\cdot; x^\top \theta_{*,i}, \sigma^2)$. This allows bounding the Bayes regret of dTS. For practice, the posterior expressions are used to motivate efficient approximations for the general case in (1) as we show in Section 3.2.

The reward distribution is parameterized as a generalized linear model (GLM) [McCullagh and Nelder, 1989], allowing for non-linear rewards. Thus, we need posterior approximation despite linearity in score functions. Since this non-linearity arises solely from the reward distribution, we approximate it by a Gaussian and propagate this approximation to the latent parameters. This results in efficient posterior approximations that are exact when the reward function is Gaussian (a special case of the GLM model). Specifically, the reward distribution $P(\cdot \mid x; \theta)$ is an exponential family distribution with a mean function denoted by $g$. Then, we approximate the corresponding likelihood as $\mathbb{P}(H_{t,i} \mid \theta_{*,i} = \theta) \approx \mathcal{N}(\theta; \hat{B}_{t,i}, \hat{G}_{t,i}^{-1})$, where $\hat{B}_{t,i}$ and $\hat{G}_{t,i}$ are the maximum likelihood estimate (MLE) and the Hessian of the negative log-likelihood, respectively, and they are defined as

$$\hat{B}_{t,i} = \arg\max_{\theta \in \mathbb{R}^d} \log \mathbb{P}(H_{t,i} \mid \theta_{*,i} = \theta), \qquad \hat{G}_{t,i} = \sum_{k \in S_{t,i}} \dot{g}(X_k^\top \hat{B}_{t,i}) X_k X_k^\top. \tag{7}$$

where $S_{t,i} = \{\ell \in [t-1] : A_\ell = i\}$ represents the rounds where the agent takes action $i$ up to round $t$. This simple approximation makes all posteriors Gaussian. Specifically, the conditional action-posterior is Gaussian and is given by $P_{t,i}(\cdot \mid \psi_1) = \mathcal{N}(\cdot; \hat{\mu}_{t,i}, \hat{\Sigma}_{t,i})$, where $\hat{\mu}_{t,i}$ and $\hat{\Sigma}_{t,i}$ are computed using $\hat{B}_{t,i}$ and $\hat{G}_{t,i}$ in (7). Moreover, for $\ell \in [L-1]$, the $\ell$-th conditional latent-posterior is also Gaussian, $Q_{t,\ell}(\cdot \mid \psi_{\ell+1}) = \mathcal{N}(\cdot; \bar{\mu}_{t,\ell}, \bar{\Sigma}_{t,\ell})$, where $\bar{\mu}_{t,\ell}$ and $\bar{\Sigma}_{t,\ell}$ are computed recursively. The recursion starts with $\bar{\mu}_{t,1}$ and $\bar{\Sigma}_{t,1}$, which are calculated using $\hat{B}_{t,i}$ and $\hat{G}_{t,i}$ in (7). Full expressions are provided in Appendix B.1. The only approximation made is $\mathbb{P}(H_{t,i} \mid \theta_{*,i} = \theta) \approx \mathcal{N}(\theta; \hat{B}_{t,i}, \hat{G}_{t,i}^{-1})$, and we propagated it to latent posteriors. Thus, these posterior approximations become exact when the reward distribution follows a linear-Gaussian model, $P(\cdot \mid x; \theta_{*,a}) = \mathcal{N}(\cdot; x^\top \theta_{*,a}, \sigma^2)$.

## 3.2 Non-linear diffusion model

After deriving the posteriors for linear score functions, we return to the general model in (1). Approximation is needed since both the score functions and rewards can be non-linear. To avoid computational challenges, we use a simple and intuitive approximation, where all posteriors $P_{t,i}$ and $Q_{t,\ell}$ are approximated by Gaussians that are computed recursively. First, the conditional action-posterior is approximated by a Gaussian distribution as $P_{t,i}(\cdot \mid \psi_1) = \mathcal{N}(\cdot; \hat{\mu}_{t,i}, \hat{\Sigma}_{t,i})$, where

$$\hat{\Sigma}_{t,i}^{-1} = \Sigma_1^{-1} + \hat{G}_{t,i} \qquad\qquad \hat{\mu}_{t,i} = \hat{\Sigma}_{t,i}(\Sigma_1^{-1} f_1(\psi_1) + \hat{G}_{t,i} \hat{B}_{t,i}). \tag{8}$$

In the absence of samples, $G_{t,i} = 0_{d \times d}$. Thus, the approximate action posterior in (8) matches precisely the term $\mathcal{N}(f_1(\psi_1), \Sigma_1)$ in the diffusion prior (1). Moreover, as more data is accumulated, $G_{t,i}$ increases, and the influence of the prior diminishes as $\hat{G}_{t,i} \hat{B}_{t,i}$ will dominate the prior term $\Sigma_1^{-1} f_1(\psi_1)$. Similarly, for $\ell \in [L]/\{1\}$, the $\ell - 1$-th conditional latent-posterior is approximated by

a Gaussian distribution as $Q_{t,\ell-1}(\cdot \mid \psi_\ell) = \mathcal{N}(\bar{\mu}_{t,\ell-1}, \bar{\Sigma}_{t,\ell-1})$, where

$$\bar{\Sigma}_{t,\ell-1}^{-1} = \Sigma_\ell^{-1} + \bar{G}_{t,\ell-1}, \qquad\qquad \bar{\mu}_{t,\ell-1} = \bar{\Sigma}_{t,\ell-1}\big(\Sigma_\ell^{-1} f_\ell(\psi_\ell) + \bar{B}_{t,\ell-1}\big), \qquad (9)$$

and the $L$-th latent-posterior is $Q_{t,L}(\cdot) = \mathcal{N}(\bar{\mu}_{t,L}, \bar{\Sigma}_{t,L})$,

$$\bar{\Sigma}_{t,L}^{-1} = \Sigma_{L+1}^{-1} + \bar{G}_{t,L}, \qquad\qquad \bar{\mu}_{t,L} = \bar{\Sigma}_{t,L}\bar{B}_{t,L}. \qquad (10)$$

Here, $\bar{G}_{t,\ell}$ and $\bar{B}_{t,\ell}$ for $\ell \in [L]$ are computed recursively. The basis of the recursion are

$$\bar{G}_{t,1} = \sum_{i=1}^{K}\big(\Sigma_1^{-1} - \Sigma_1^{-1}\hat{\Sigma}_{t,i}\Sigma_1^{-1}\big), \qquad\qquad \bar{B}_{t,1} = \Sigma_1^{-1}\sum_{i=1}^{K}\hat{\Sigma}_{t,i}\hat{G}_{t,i}\hat{B}_{t,i}. \qquad (11)$$

Then, the recursive step for $\ell \in [L]/\{1\}$ is,

$$\bar{G}_{t,\ell} = \Sigma_\ell^{-1} - \Sigma_\ell^{-1}\bar{\Sigma}_{t,\ell-1}\Sigma_\ell^{-1}, \qquad\qquad \bar{B}_{t,\ell} = \Sigma_\ell^{-1}\bar{\Sigma}_{t,\ell-1}\bar{B}_{t,\ell-1}. \qquad (12)$$

Similarly, in the absence of samples, $Q_{t,\ell-1}$ in (9) precisely matches the term $\mathcal{N}(f_\ell(\psi_1), \Sigma_\ell)$ in the diffusion prior (1). As more data is accumulated, the influence of this prior diminishes. Therefore, this approximation retains a key attribute of exact posteriors: they match the prior when there is no data, and the prior's effect diminishes as data accumulates.

# 4 Analysis

We analyze dTS under the linear diffusion model in Section 3.1 with linear rewards $P(\cdot \mid x; \theta_{*,a}) = \mathcal{N}(\cdot; x^\top \theta_{*,a}, \sigma^2)$. This assumption leads to a structure with $L$ layers of linear Gaussian relationships, allowing for theory inspired by linear bandits [Agrawal and Goyal, 2013a, Abbasi-Yadkori et al., 2011]. However, proofs are not the same, and technical challenges remain (explained in Appendix D).

Although our result holds for milder assumptions, we make some simplifications for clarity and interpretability. We assume that **(A1)** Contexts satisfy $\|X_t\|_2^2 = 1$ for any $t \in [n]$. **(A2)** Mixing matrices and covariances satisfy $\lambda_1(W_\ell^\top W_\ell) = 1$ for any $\ell \in [L]$ and $\Sigma_\ell = \sigma_\ell^2 I_d$ for any $\ell \in [L+1]$. Note that **(A1)** can be relaxed to any contexts $X_t$ with bounded norms $\|X_t\|_2$. Also, **(A2)** can be relaxed to positive definite covariances $\Sigma_\ell$ and arbitrary mixing matrices $W_\ell$. In this section, we write $\tilde{\mathcal{O}}$ for the big-O notation up to polylogarithmic factors. We start by stating our bound for dTS.

**Theorem 4.1.** *Let $\sigma_{\text{MAX}}^2 = \max_{\ell \in [L+1]} 1 + \frac{\sigma_\ell^2}{\sigma^2}$. For any $\delta \in (0,1)$, the Bayes regret of* dTS *under Section 3.1 with linear rewards, **(A1)** and **(A2)** is bounded as*

$$\mathcal{BR}(n) \leq \sqrt{2n\big(\mathcal{R}^{\text{ACT}}(n) + \sum_{\ell=1}^{L} \mathcal{R}_\ell^{\text{LAT}}\big)\log(1/\delta)} + cn\delta, \text{ with } c > 0 \text{ is constant and,} \qquad (13)$$

$$\mathcal{R}^{\text{ACT}}(n) = c_0 dK \log\big(1 + \tfrac{n\sigma_1^2}{d}\big),\ c_0 = \tfrac{\sigma_1^2}{\log(1+\sigma_1^2)}, \quad \mathcal{R}_\ell^{\text{LAT}} = c_\ell d \log\big(1 + \tfrac{\sigma_{\ell+1}^2}{\sigma_\ell^2}\big),\ c_\ell = \tfrac{\sigma_{\ell+1}^2 \sigma_{\text{MAX}}^{2\ell}}{\log(1+\sigma_{\ell+1}^2)},$$

(13) holds for any $\delta \in (0,1)$. In particular, the term $cn\delta$ is constant when $\delta = 1/n$. Then, the bound is $\tilde{\mathcal{O}}(\sqrt{n})$, and this dependence on the horizon $n$ aligns with prior Bayes regret bounds. The bound comprises $L + 1$ main terms, $\mathcal{R}^{\text{ACT}}(n)$ and $\mathcal{R}_\ell^{\text{LAT}}$ for $\ell \in [L]$. First, $\mathcal{R}^{\text{ACT}}(n)$ relates to action parameters learning, conforming to a standard form [Lu and Van Roy, 2019]. Similarly, $\mathcal{R}_\ell^{\text{LAT}}$ is associated with learning the $\ell$-th latent parameter. Roughly speaking, our bound captures that our problem can be seen as $L + 1$ sequential linear bandit instances stacked upon each other.

**Technical contributions.** dTS uses hierarchical sampling. Thus the marginal posterior distribution of $\theta_{*,i} \mid H_t$ is not explicitly defined. The first contribution is deriving $\theta_{*,i} \mid H_t$ using the total covariance decomposition combined with an induction proof, as our posteriors in Section 3.1 were derived recursively. Unlike standard analyses where the posterior distribution of $\theta_{*,i} \mid H_t$ is predetermined due to the absence of latent parameters, our method necessitates this recursive total covariance decomposition. Moreover, in standard proofs, we need to quantify the increase in posterior precision for the action taken $A_t$ in each round $t \in [n]$. However, in dTS, our analysis extends beyond this. We not only quantify the posterior information gain for the taken action but also for every latent parameter, since they are also learned. To elaborate, we use the recursive formulas in Section 3.1 that connect the posterior covariance of each latent parameter $\psi_{*,\ell}$ with the covariance of the posterior action parameters $\theta_{*,i}$. This allows us to propagate the information gain associated with the action

taken in round $A_t$ to all latent parameters $\psi_{*,\ell}$, for $\ell \in [L]$ by induction. Finally, we carefully bound the resulting terms so that the constants reflect the parameters of the linear diffusion model. More technical details are provided in Appendix D.

To include more structure, we propose the *sparsity* assumption **(A3)** $W_\ell = (\bar{W}_\ell, 0_{d,d-d_\ell})$, where $\bar{W}_\ell \in \mathbb{R}^{d \times d_\ell}$ for any $\ell \in [L]$. Note that **(A3)** is not an assumption when $d_\ell = d$ for any $\ell \in [L]$. Notably, **(A3)** incorporates a plausible structural characteristic that a diffusion model could capture.

**Proposition 4.2** (Sparsity). *Let* $\sigma_{\text{MAX}}^2 = \max_{\ell \in [L+1]} 1 + \frac{\sigma_\ell^2}{\sigma^2}$. *For any* $\delta \in (0,1)$, *the Bayes regret of* dTS *under Section 3.1 with linear rewards,* **(A1)**, **(A2)** *and* **(A3)** *is bounded as*

$$\mathcal{BR}(n) \leq \sqrt{2n\big(\mathcal{R}^{\text{ACT}}(n) + \sum_{\ell=1}^{L} \tilde{\mathcal{R}}_\ell^{\text{LAT}}\big)\log(1/\delta)} + cn\delta, \text{ with } c > 0 \text{ is constant,} \quad (14)$$

$$\mathcal{R}^{\text{ACT}}(n) = c_0 dK \log\big(1 + \tfrac{n\sigma_1^2}{d}\big), c_0 = \tfrac{\sigma_1^2}{\log(1+\sigma_1^2)}, \quad \tilde{\mathcal{R}}_\ell^{\text{LAT}} = c_\ell d_\ell \log\big(1 + \tfrac{\sigma_{\ell+1}^2}{\sigma_\ell^2}\big), c_\ell = \tfrac{\sigma_{\ell+1}^2 \sigma_{\text{MAX}}^{2\ell}}{\log(1+\sigma_{\ell+1}^2)}.$$

From Proposition 4.2, our bounds scales as $\mathcal{BR}(n) = \tilde{\mathcal{O}}\Big(\sqrt{n(dK\sigma_1^2 + \sum_{\ell=1}^{L} d_\ell \sigma_{\ell+1}^2 \sigma_{\text{MAX}}^{2\ell})}\Big)$. The Bayes regret bound has a clear interpretation: if the true environment parameters are drawn from the prior, then the expected regret of an algorithm stays below that bound. Consequently, a less informative prior (such as high variance) leads to a more challenging problem and thus a higher bound. Then, smaller values of $K$, $L$, $d$ or $d_\ell$ translate to fewer parameters to learn, leading to lower regret. The regret also decreases when the initial variances $\sigma_\ell^2$ decrease. These dependencies are common in Bayesian analysis, and empirical results match them. The reader might question the dependence of our bound on both $L$ and $K$. We will address this next.

**Why the bound increases with $K$?** This arises due to our conditional learning of $\theta_{*,i}$ given $\psi_{*,1}$. Rather than assuming deterministic linearity, $\theta_{*,i} = W_1 \psi_{*,1}$, we account for stochasticity by modeling $\theta_{*,i} \sim \mathcal{N}(W_1 \psi_{*,1}, \sigma_1^2 I_d)$. This makes dTS robust to misspecification scenarios where $\theta_{*,i}$ is not perfectly linear with respect to $\psi_{*,1}$, at the cost of additional learning of $\theta_{*,i} \mid \psi_{*,1}$. If we were to assume deterministic linearity ($\sigma_1 = 0$), our regret bound would scale with $L$ only.

**Why the bound increases with $L$?** This is because increasing the number of layers $L$ adds more initial uncertainty due to the additional covariance introduced by the extra layers. However, this does not imply that we should always use $L = 1$ (the minimum possible $L$). While a higher $L$ complicates online learning and increases regret bound, it also enables the capture of a more complex prior distribution through offline pre-training of the diffusion model. Thus, a trade-off exists in practice. A smaller $L$ results in faster computation and easier learning for dTS, but the learned prior might deviate from reality, potentially violating the "true prior assumption" used to derive the regret bound. On the other hand, a larger $L$ allows for better modeling of complex action distributions, producing a prior that more accurately reflects reality and strengthens the validity of the bound.

## 4.1 Discussion

**Computational benefits.** Action correlations prompt an intuitive approach: marginalize all latent parameters and maintain a joint posterior of $(\theta_{*,i})_{i \in [K]} \mid H_t$. Unfortunately, this is computationally inefficient for large action spaces. To illustrate, suppose that all posteriors are multivariate Gaussians (Section 3.1). Then maintaining the joint posterior $(\theta_{*,i})_{i \in [K]} \mid H_t$ necessitates converting and storing its $dK \times dK$-dimensional covariance matrix. Then the time and space complexities are $\mathcal{O}(K^3 d^3)$ and $\mathcal{O}(K^2 d^2)$. In contrast, the time and space complexities of dTS are $\mathcal{O}\big((L + K)d^3\big)$ and $\mathcal{O}\big((L + K)d^2\big)$. This is because dTS requires converting and storing $L + K$ covariance matrices, each being $d \times d$-dimensional. The improvement is huge when $K \gg L$, which is common in practice. Certainly, a more straightforward way to enhance computational efficiency is to discard latent parameters and maintain $K$ individual posteriors, each relating to an action parameter $\theta_{*,i} \in \mathbb{R}^d$ (LinTS). This improves time and space complexity to $\mathcal{O}(Kd^3)$ and $\mathcal{O}(Kd^2)$, respectively. However, LinTS maintains independent posteriors and fails to capture the correlations among actions; it only models $\theta_{*,i} \mid H_{t,i}$ rather than $\theta_{*,i} \mid H_t$ as done by dTS. Consequently, LinTS incurs higher regret due to the information loss caused by unused interactions of similar actions. Our regret bound and empirical results reflect this aspect.

**Statistical benefits.** We do not provide a matching lower bound. The only Bayesian lower bound that we know of is $\Omega(\log^2(n))$ for a much simpler $K$-armed bandit [Lai, 1987, Theorem 3]. All

seminal works on Bayesian bandits do not match it and providing such lower bounds on Bayes regret is still relatively unexplored (even in standard settings) compared to the frequentist one. Therefore, we argue that our bound reflects the overall structure of the problem by comparing dTS to algorithms that only partially use the structure or do not use it at all as follows.

The linear diffusion model in Section 3.1 can be transformed into a Bayesian linear model (LinTS) by marginalizing out the latent parameters; in which case the prior on action parameters becomes $\theta_{*,i} \sim \mathcal{N}(0, \Sigma)$, with the $\theta_{*,i}$ being not necessarily independent, and $\Sigma$ is the marginal initial covariance of action parameters and it writes $\Sigma = \sigma_1^2 I_d + \sum_{\ell=1}^{L} \sigma_{\ell+1}^2 \mathrm{B}_\ell \mathrm{B}_\ell^\top$ with $\mathrm{B}_\ell = \prod_{k=1}^{\ell} \mathrm{W}_k$. Then, it is tempting to directly apply LinTS to solve our problem. This approach will induce higher regret because the additional uncertainty of the latent parameters is accounted for in $\Sigma$ despite integrating them. This causes the *marginal* action uncertainty $\Sigma$ to be much higher than the *conditional* action uncertainty $\sigma_1^2 I_d$ in (3.1), since we have $\Sigma = \sigma_1^2 I_d + \sum_{\ell=1}^{L} \sigma_{\ell+1}^2 \mathrm{B}_\ell \mathrm{B}_\ell^\top \succcurlyeq \sigma_1^2 I_d$. This discrepancy leads to higher regret, especially when $K$ is large. This is due to LinTS needing to learn $K$ independent $d$-dimensional parameters, each with a considerably higher initial covariance $\Sigma$. This is also reflected by our regret bound. To simply comparisons, suppose that $\sigma \geq \max_{\ell \in [L+1]} \sigma_\ell$ so that $\sigma_{\text{MAX}}^2 \leq 2$. Then the regret bounds of dTS (where we bound $\sigma_{\text{MAX}}^{2\ell}$ by $2^\ell$) and LinTS read

$$\texttt{dTS} : \tilde{\mathcal{O}}\big(\sqrt{n(dK\sigma_1^2 + \sum_{\ell=1}^{L} d_\ell \sigma_{\ell+1}^2 2^\ell)}\big), \qquad \texttt{LinTS} : \tilde{\mathcal{O}}\big(\sqrt{ndK(\sigma_1^2 + \sum_{\ell=1}^{L} \sigma_{\ell+1}^2)}\big).$$

Then regret improvements are captured by the variances $\sigma_\ell$ and the sparsity dimensions $d_\ell$, and we proceed to illustrate this through the following scenarios.

**(I) Decreasing variances.** Assume that $\sigma_\ell = 2^\ell$ for any $\ell \in [L+1]$. Then, the regrets become

$$\texttt{dTS} : \tilde{\mathcal{O}}\big(\sqrt{n(dK + \sum_{\ell=1}^{L} d_\ell 4^\ell))}\big), \qquad \texttt{LinTS} : \tilde{\mathcal{O}}\big(\sqrt{ndK2^L}\big)$$

Now to see the order of gain, assume the problem is high-dimensional ($d \gg 1$), and set $L = \log_2(d)$ and $d_\ell = \lfloor \frac{d}{2^\ell} \rfloor$. Then the regret of dTS becomes $\tilde{\mathcal{O}}\big(\sqrt{nd(K+L)}\big)$, and hence the multiplicative factor $2^L$ in LinTS is removed and replaced with a smaller additive factor $L$.

**(II) Constant variances.** Assume that $\sigma_\ell = 1$ for any $\ell \in [L+1]$. Then, the regrets become

$$\texttt{dTS} : \tilde{\mathcal{O}}\big(\sqrt{n(dK + \sum_{\ell=1}^{L} d_\ell 2^\ell))}\big), \qquad \texttt{LinTS} : \tilde{\mathcal{O}}\big(\sqrt{ndKL}\big)$$

Similarly, let $L = \log_2(d)$, and $d_\ell = \lfloor \frac{d}{2^\ell} \rfloor$. Then dTS's regret is $\tilde{\mathcal{O}}\big(\sqrt{nd(K+L)}\big)$. Thus the multiplicative factor $L$ in LinTS is removed and replaced with the additive factor $L$. By comparing this to **(I)**, the gain with decreasing variances is greater than with constant ones. In general, diffusion models use decreasing variances [Ho et al., 2020] and hence we expect great gains in practice. All observed improvements in this section could become even more pronounced when employing non-linear diffusion models. In our current analysis, we used linear diffusion models, and yet we can already discern substantial differences. Moreover, under non-linear diffusion (1), the latent parameters cannot be analytically marginalized, making LinTS with exact marginalization inapplicable. Finally, Appendix D.7 provide an additional comparison and connection to hierarchies with two levels.

**Large action space aspect.** dTS's regret bound scales with $K\sigma_1^2$ instead of $K \sum_\ell \sigma_\ell^2$, particularly beneficial when $\sigma_1$ is small, as often seen in diffusion models. Our regret bound and experiments show that dTS outperforms LinTS more distinctly when the action space becomes larger. Prior studies [Foster et al., 2020, Xu and Zeevi, 2020, Zhu et al., 2022] proposed bandit algorithms that do not scale with $K$. However, our setting differs significantly from theirs, explaining our inherent dependency on $K$ when $\sigma_1 > 0$. Precisely, they assume a reward function of $r(x, i; \theta_*) = \phi(x, i)^\top \theta_*$, with a shared $\theta_* \in \mathbb{R}^d$ and a known mapping $\phi$. In contrast, we consider $r(x, i; \theta_*) = x^\top \theta_{*,i}$, with $\theta_* = (\theta_{*,i})_{i \in [K]} \in \mathbb{R}^{dK}$, requiring the learning of $K$ separate $d$-dimensional action parameters. In their setting, with the availability of $\phi$, the regret of dTS would similarly be independent of $K$. However, obtaining such a mapping $\phi$ can be challenging as it needs to encapsulate complex context-action dependencies. Notably, our setting reflects a common practical scenario, such as in recommendation systems where each product is often represented by its unique embedding.

## 5 Experiments

We evaluate dTS using synthetic data, to validate our theory and test dTS in large action spaces. We omit semi-synthetic data [Riquelme et al., 2018] as they often result in small action spaces. This

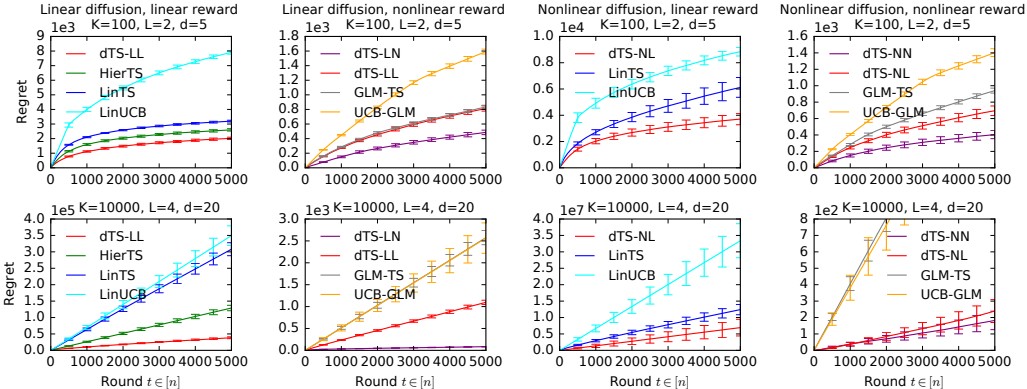

Figure 2: Regret of dTS with varying diffusion and reward models and varying parameters $d, K, L$.

choice is further justified by the fact that Hsieh et al. [2023] has already demonstrated the advantages of diffusion models in multi-armed bandits using such data, without theoretical guarantees.

## 5.1 Settings and baselines

We run 50 random simulations and plot the average regret with its standard error. We consider both linear and non-linear rewards. The distribution of linear rewards is $P(\cdot \mid x; \theta_a) = \mathcal{N}(x^\top \theta_a, \sigma^2)$ with $\sigma = 1$. The non-linear rewards are binary and generated from $P(\cdot \mid x; \theta_a) = \mathrm{Ber}(g(x^\top \theta_a))$, where $g$ is the sigmoid function. The covariances are $\Sigma_\ell = I_d$, and the context $X_t$ is uniformly drawn from $[-1, 1]^d$. We vary $d \in \{5, 20\}$, $L \in \{2, 4\}$ and $K \in \{10^2, 10^4\}$. We set the horizon $n = 5000$.

**Linear diffusion.** We consider the linear diffusion model in (3.1) where score functions are linear as $f_\ell(\psi) = W_\ell \psi$ where $W_\ell$ are uniformly drawn from $[-1, 1]^{d \times d}$. To introduce sparsity, we zero out the last $d_\ell$ columns of $W_\ell$, resulting in $W_\ell = (\bar{W}_\ell, 0_{d, d - d_\ell})$, where $(d_1, d_2) = (5, 2)$ when $d = 5$ and $L = 2$ and $(d_1, d_2, d_3, d_4) = (20, 10, 5, 2)$ when $d = 20$ and $L = 4$.

**Non-linear diffusion.** We consider the general diffusion model in (1) with score functions $f_\ell$ defined by two-layer neural networks with random weights in $[-1, 1]$, ReLU activation, and a hidden layer dimension of $h = 20$ when $d = 5$ and $h = 60$ when $d = 20$.

**Baselines.** When rewards are linear, we use LinUCB [Abbasi-Yadkori et al., 2011], LinTS [Agrawal and Goyal, 2013a], and HierTS [Hong et al., 2022b] that marginalizes out all latent parameters except $\psi_{*,L}$. This corresponds to HierTS-1 in Appendix D.7. When rewards are non-linear, we include UCB-GLM [Li et al., 2017], and GLM-TS [Chapelle and Li, 2012]. GLM-UCB [Filippi et al., 2010] induced high regret while HierTS was designed for linear rewards only and thus both are not included. We name dTS for each setting as dTS-dr, where the suffix d indicates the type of diffusion; L for linear and N for non-linear. The suffix r indicates the type of rewards; L for linear and N for non-linear. For instance, dTS-LL signifies dTS in linear diffusion (Section 3.1) with linear rewards.

## 5.2 Results and interpretations

Results are shown in Fig. 2 and we make the following observations:

**1) dTS has better performance.** dTS outperforms the baselines. First, when both the diffusion and rewards are linear, dTS-LL consistently outperforms all baselines that disregard the latent structure (LinTS and LinUCB) or incorporate it only partially (HierTS). Second, when the diffusion is linear and rewards are non-linear, dTS-LN surpasses all baselines. Third, when the diffusion is non-linear and rewards are linear, dTS-NL demonstrates significant performance gains compared to both LinTS and LinUCB. With non-linear diffusion and rewards, dTS-NN surpasses both GLM-TS and UCB-GLM.

**2) Latent diffusion structure may be more important than the reward distribution.** When rewards are non-linear (second and fourth columns in Fig. 2), we included variants of dTS that use the correct diffusion prior but the wrong reward distribution, employing linear-Gaussian instead of logistic-Bernoulli (dTS-LL in the second column and dTS-NL in the fourth column). In both cases, despite the misspecification of the reward distribution, these variants outperform models that use the correct reward distribution but neglect the latent diffusion structure, such as GLM-TS and UCB-GLM.

This underscores the significance of accounting for the latent structure, which can sometimes be more crucial than having an accurate reward distribution. Also, the performance gap between `dTS-NL` (non-linear diffusion) and `GLM-TS` and `UCB-GLM` is even more pronounced compared to the gap between `dTS-LL` (linear diffusion) and these baselines, possibly due to the increased complexity of the latent structure, in the non-linear diffusion, overshadowing the impact of the reward model itself.

**3) Prior misspecification (Fig. 3).** We consider a scenario where the prior used by `dTS` does not match the true prior. To simulate this, we use our setting with linear diffusion and rewards above, but the true parameters $W_\ell$ and $\Sigma_\ell$ are replaced by misspecified parameters $W_\ell + \epsilon_1$ and $\Sigma_\ell + \epsilon_2$. Here, $\epsilon_1$ and $\epsilon_2$ are sampled uniformly from $[v, v+0.5]^{d \times d}$, with $v$ controlling the level of misspecification. The higher the value of $v$, the greater the misspecification. We vary $v \in \{0.5, 1, 1.5\}$ and analyze its impact on `dTS`'s performance. For comparison, we include the well-specified `dTS-LL` and the most competitive baseline, `HierTS`. Results are shown in Fig. 3. As expected, `dTS`'s performance decreases with increasing misspecification. However, even

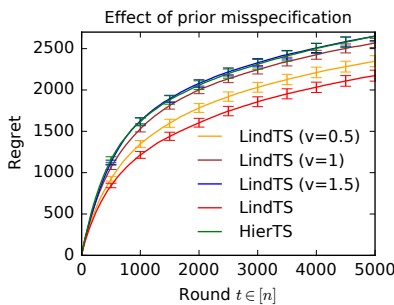

Figure 3: Prior misspecification effect.

with misspecification, `dTS` outperforms the most competitive baseline, except when $v = 1.5$, where their performances are comparable. Note that the entries of the true parameters $W_\ell$ and $\Sigma_\ell$ are smaller than 1, so values of $v \in \{0.5, 1, 1.5\}$ can lead to significant parameter misspecification. Yet, the performance of `dTS` with misspecified prior parameters remains favorable, suggesting that even an imperfect pre-trained diffusion model can be beneficial when used as prior.

**4) Regret scaling with $K$, $d$ and $L$ matches our theory (Fig. 4).** We verify the impact of the number of actions $K$, the context dimension $d$, and the diffusion depth $L$ on the regret of `dTS`. We maintain the same experimental setup with linear diffusion and rewards, for which we have derived a Bayes regret upper bound. In Fig. 4, we plot the regret of `dTS-LL` across varying values of these parameters: $K \in \{10, 100, 500, 1000\}$, $d \in \{5, 10, 15, 20\}$, and $L \in \{2, 4, 5, 6\}$. As anticipated and aligned with our theory, the empirical regret increases as the values of $K$, $d$, or $L$ grow. This trend arises because larger values of $K$, $d$, or $L$ result in problem instances that are more challenging to learn, consequently leading to higher regret.

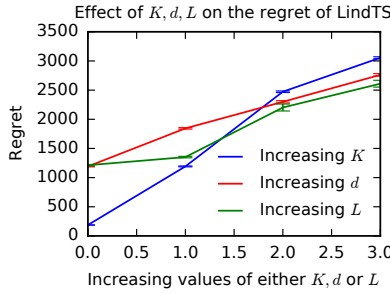

Figure 4: `dTS-LL`'s regret scaling.

**5) Performance gap between `dTS` and `LinTS` widens as $K$ increases (Fig. 5).** To showcase `dTS`'s improved scalability to larger action spaces, we examine its performance across a range of $K$ values, from 10 to $50,000$, in our setting with linear diffusion and rewards. Fig. 5 reports the final cumulative regret for varying values of $K$ for both `dTS-LL` and `LinTS`, observing that the gap in performance becomes larger as $K$ increases.

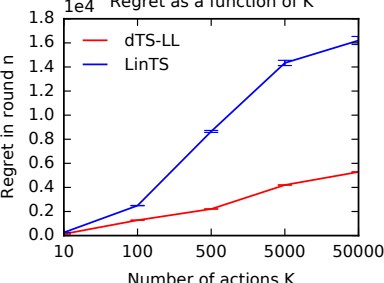

Figure 5: Regret of `dTS-LL` and `LinTS` with varying $K$.

## 6 Conclusion

Grappling with large action spaces in contextual bandits is challenging. Recognizing this, we focused on structured problems where action parameters are sampled from a diffusion model; upon which we built diffusion Thompson sampling (`dTS`). We developed both theoretical and algorithmic foundations for `dTS` in numerous practical settings. We identified several directions for future work. Exploring other approximations for non-linear diffusion models, both empirically and theoretically. From a theoretical perspective, future research could explore the advantages of non-linear diffusion models by deriving their Bayes regret bounds, akin to our analysis in Section 4. Empirically, investigating our and other approximations in complex tasks would be interesting. Additionally, exploring the extension of this work to offline (or off-policy) learning in contextual bandits [Swaminathan and Joachims, 2015, Aouali et al., 2023a] represents a promising avenue for future research.

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

## Supplementary materials

**Notation.** For any positive integer $n$, we define $[n] = \{1, 2, ..., n\}$. Let $v_1, \ldots, v_n \in \mathbb{R}^d$ be $n$ vectors, $(v_i)_{i \in [n]} \in \mathbb{R}^{nd}$ is the $nd$-dimensional vector obtained by concatenating $v_1, \ldots, v_n$. For any matrix $A \in \mathbb{R}^{d \times d}$, $\lambda_1(A)$ and $\lambda_d(A)$ denote the maximum and minimum eigenvalues of A, respectively. Finally, we write $\tilde{\mathcal{O}}$ for the big-O notation up to polylogarithmic factors.

## A Extended related work

**Thompson sampling (TS)** operates within the Bayesian framework and it involves specifying a prior/likelihood model. In each round, the agent samples unknown model parameters from the current posterior distribution. The chosen action is the one that maximizes the resulting reward. TS is naturally randomized, particularly simple to implement, and has highly competitive empirical performance in both simulated and real-world problems [Russo and Van Roy, 2014, Chapelle and Li, 2012]. Regret guarantees for the TS heuristic remained open for decades even for simple models. Recently, however, significant progress has been made. For standard multi-armed bandits, TS is optimal in the Beta-Bernoulli model [Kaufmann et al., 2012, Agrawal and Goyal, 2013b], Gaussian-Gaussian model [Agrawal and Goyal, 2013b], and in the exponential family using Jeffrey's prior [Korda et al., 2013]. For linear bandits, TS is nearly-optimal [Russo and Van Roy, 2014, Agrawal and Goyal, 2017, Abeille and Lazaric, 2017]. In this work, we build TS upon complex diffusion priors and analyze the resulting Bayes regret [Russo and Van Roy, 2014] in the linear contextual bandit setting.

**Decision-making with diffusion models** gained attention recently, especially in offline learning [Ajay et al., 2022, Janner et al., 2022, Wang et al., 2022]. However, their application in online learning was only examined by Hsieh et al. [2023], which focused on meta-learning in multi-armed bandits without theoretical guarantees. In this work, we expand the scope of Hsieh et al. [2023] to encompass the broader contextual bandit framework. In particular, we provide theoretical analysis for linear instances, effectively capturing the advantages of using diffusion models as priors in contextual Thompson sampling. These linear cases are particularly captivating due to closed-form posteriors, enabling both theoretical analysis and computational efficiency; an important practical consideration.

**Hierarchical Bayesian bandits** [Bastani et al., 2019, Kveton et al., 2021, Basu et al., 2021, Simchowitz et al., 2021, Wan et al., 2021, Hong et al., 2022b, Peleg et al., 2022, Wan et al., 2022, Aouali et al., 2023b] applied TS to simple graphical models, wherein action parameters are generally sampled from a Gaussian distribution centered at a single latent parameter. These works mostly span meta- and multi-task learning for multi-armed bandits, except in cases such as Aouali et al. [2023b], Hong et al. [2022a] that consider the contextual bandit setting. Precisely, Aouali et al. [2023b] assume that action parameters are sampled from a Gaussian distribution centered at a linear mixture of multiple latent parameters. On the other hand, Hong et al. [2022a] applied TS to a graphical model represented by a tree. Our work can be seen as an extension of all these works to much more complex graphical models, for which both theoretical and algorithmic foundations are developed. Note that the settings in most of these works can be recovered with specific choices of the diffusion depth $L$ and functions $f_\ell$. This attests to the modeling power of dTS.

**Approximate Thompson sampling** is a major problem in the Bayesian inference literature. This is because most posterior distributions are intractable, and thus practitioners must resort to sophisticated computational techniques such as Markov chain Monte Carlo [Kruschke, 2010]. Prior works [Riquelme et al., 2018, Chapelle and Li, 2012, Kveton et al., 2020] highlight the favorable empirical performance of approximate Thompson sampling. Particularly, [Kveton et al., 2020] provide theoretical guarantees for Thompson sampling when using the Laplace approximation in generalized linear bandits (GLB). In our context, we incorporate approximate sampling when the reward exhibits non-linearity. While our approximation does not come with formal guarantees, it enjoys strong practical performance. An in-depth analysis of this approximation is left as a direction for future works. Similarly, approximating the posterior distribution when the diffusion model is non-linear as well as analyzing it is an interesting direction of future works.

**Bandits with underlying structure** also align with our work, where we assume a structured relationship among actions, captured by a diffusion model. In latent bandits [Maillard and Mannor, 2014, Hong et al., 2020], a single latent variable indexes multiple candidate models. Within structured

finite-armed bandits [Lattimore and Munos, 2014, Gupta et al., 2018], each action is linked to a known mean function parameterized by a common latent parameter. This latent parameter is learned. TS was also applied to complex structures [Yu et al., 2020, Gopalan et al., 2014]. However, simultaneous computational and statistical efficiencies aren't guaranteed. Meta- and multi-task learning with upper confidence bound (UCB) approaches have a long history in bandits [Azar et al., 2013, Gentile et al., 2014, Deshmukh et al., 2017, Cella et al., 2020]. These, however, often adopt a frequentist perspective, analyze a stronger form of regret, and sometimes result in conservative algorithms. In contrast, our approach is Bayesian, with analysis centered on Bayes regret. Remarkably, our algorithm, dTS, performs well as analyzed without necessitating additional tuning. Finally, **Low-rank bandits** [Hu et al., 2021, Cella et al., 2022, Yang et al., 2020] also relate to our linear diffusion model when $L = 1$. Broadly, there exist two key distinctions between these prior works and the special case of our model (linear diffusion model with $L = 1$). First, they assume $\theta_{*,i} = W_1 \psi_{*,1}$, whereas we incorporate additional uncertainty in the covariance $\Sigma_1$ to account for possible misspecification as $\theta_{*,i} = \mathcal{N}(W_1 \psi_{*,1}, \Sigma_1)$. Consequently, these algorithms might suffer linear regret due to model misalignment. Second, we assume that the mixing matrix $W_1$ is available and pre-learned offline, whereas they learn it online. While this is more general, it leads to computationally expensive methods that are difficult to employ in a real-world online setting.

**Large action spaces.** Roughly speaking, the regret bound of dTS scales with $K\sigma_1^2$ rather than $K \sum_\ell \sigma_\ell^2$. This is particularly beneficial when $\sigma_1$ is small, a common scenario in diffusion models with decreasing variances. A notable case is when $\sigma_1 = 0$, where the regret becomes independent of $K$. Also, our analysis (Section 4.1) indicates that the gap in performance between dTS and LinTS becomes more pronounced when the number of action increases, highlighting dTS's suitability for large action spaces. Note that some prior works [Foster et al., 2020, Xu and Zeevi, 2020, Zhu et al., 2022] proposed bandit algorithms that do not scale with $K$. However, our setting differs significantly from theirs, explaining our inherent dependency on $K$ when $\sigma_1 > 0$. Precisely, they assume a reward function of $r(x, i) = \phi(x, i)^\top \theta_*$, with a shared $\theta_* \in \mathbb{R}^d$ across actions and a known mapping $\phi$. In contrast, we consider $r(x, i) = x^\top \theta_{*,i}$, requiring the learning of $K$ separate $d$-dimensional action parameters. In their setting, with the availability of $\phi$, the regret of dTS would similarly be independent of $K$. However, obtaining such a mapping $\phi$ can be challenging as it needs to encapsulate complex context-action dependencies. Notably, our setting reflects a common practical scenario, such as in recommendation systems where each product is often represented by its embedding. In summary, the dependency on $K$ is more related to our setting than the method itself, and dTS would scale with $d$ only in their setting. Note that dTS is both computationally and statistically efficient (Section 4.1). This becomes particularly notable in large action spaces. Our empirical results in Fig. 2, notably with $K = 10^4$, demonstrate that dTS significantly outperforms the baselines. More importantly, the performance gap between dTS and these baselines is larger when the number of actions ($K$) increases, highlighting the improved scalability of dTS to large action spaces.

# B  Posterior derivations for linear diffusion models

Here, we assume the score functions $f_\ell$ are linear such as $f_\ell(\psi_{*,\ell}) = W_\ell \psi_{*,\ell}$ for $\ell \in [L]$, where $W_\ell \in \mathbb{R}^{d \times d}$ are *known mixing matrices*. Then, (1) becomes a linear Gaussian system (LGS) [Bishop, 2006] and can be summarized as follows

$$
\begin{aligned}
\psi_{*,L} &\sim \mathcal{N}(0, \Sigma_{L+1}), & &(15)\\
\psi_{*,\ell-1} \mid \psi_{*,\ell} &\sim \mathcal{N}(W_\ell \psi_{*,\ell}, \Sigma_\ell), & &\forall \ell \in [L]/\{1\},\\
\theta_{*,i} \mid \psi_{*,1} &\sim \mathcal{N}(W_1 \psi_{*,1}, \Sigma_1), & &\forall i \in [K],\\
Y_t \mid X_t, \theta_{*,A_t} &\sim P(\cdot \mid X_t; \theta_{*,A_t}), & &\forall t \in [n].
\end{aligned}
$$

In this section, we derive the $K + L$ posteriors $P_{t,i}$ and $Q_{t,\ell}$, for which we provide the full expressions in Appendix B.1. In our proofs, $p(x) \propto f(x)$ means that the probability density $p$ satisfies $p(x) = \frac{f(x)}{Z}$ for any $x \in \mathbb{R}^d$, where $Z$ is a normalization constant. In particular, we extensively use that if $p(x) \propto \exp[-\frac{1}{2} x^\top \Lambda x + x^\top m]$, where $\Lambda$ is positive definite. Then $p$ is the multivariate Gaussian density with covariance $\Sigma = \Lambda^{-1}$ and mean $\mu = \Sigma m$. These are standard notations and techniques to manipulate Gaussian distributions [Koller and Friedman, 2009, Chapter 7].

## B.1 Posterior expressions for linear diffusion models

Recall that we posit that the reward distribution is parameterized as a generalized linear model (GLM) [McCullagh and Nelder, 1989], allowing for non-linear rewards. As a result, despite linearity in score functions, the non-linearity in rewards makes it challenging to obtain closed-form posteriors. However, since this non-linearity arises solely from the reward distribution, we approximate it using a Gaussian distribution. This leads to efficient posterior approximations that are exact in cases where the reward function is indeed Gaussian (a special case of the GLM model). Precisely, the reward distribution $P(\cdot \mid x; \theta)$ is an exponential-family distribution. Therefore, the log-likelihoods write $\log \mathbb{P}\left(H_{t,i} \mid \theta_{*,i} = \theta\right) = \sum_{k \in S_{t,i}} Y_k X_k^\top \theta - A(X_k^\top \theta) + C(Y_k)$, where $C$ is a real function, and $A$ is a twice continuously differentiable function whose derivative is the mean function, $\dot{A} = g$. Now we let $\hat{B}_{t,i}$ and $\hat{G}_{t,i}$ be the maximum likelihood estimate (MLE) and the Hessian of the negative log-likelihood, respectively, defined as

$$\hat{B}_{t,i} = \arg\max_{\theta \in \mathbb{R}^d} \log \mathbb{P}\left(H_{t,i} \mid \theta_{*,i} = \theta\right), \qquad \hat{G}_{t,i} = \sum_{k \in S_{t,i}} \dot{g}\left(X_k^\top \hat{B}_{t,i}\right) X_k X_k^\top. \qquad (16)$$

where $S_{t,i} = \{\ell \in [t-1] : A_\ell = i\}$ are the rounds where the agent takes action $i$ up to round $t$. Then we approximation the respective likelihood as $\mathbb{P}\left(H_{t,i} \mid \theta_{*,i} = \theta\right) \approx \mathcal{N}\left(\theta; \hat{B}_{t,i}, \hat{G}_{t,i}^{-1}\right)$. This approximation makes all posteriors Gaussian. First, the conditional action-posterior reads $P_{t,i}(\cdot \mid \psi_1) = \mathcal{N}(\cdot; \hat{\mu}_{t,i}, \hat{\Sigma}_{t,i})$,

$$\hat{\Sigma}_{t,i}^{-1} = \Sigma_1^{-1} + \hat{G}_{t,i} \qquad \hat{\mu}_{t,i} = \hat{\Sigma}_{t,i}\left(\Sigma_1^{-1} W_1 \psi_1 + \hat{G}_{t,i} \hat{B}_{t,i}\right). \qquad (17)$$

For $\ell \in [L]/\{1\}$, the $\ell - 1$-th conditional latent-posterior is $Q_{t,\ell-1}(\cdot \mid \psi_\ell) = \mathcal{N}(\bar{\mu}_{t,\ell-1}, \bar{\Sigma}_{t,\ell-1})$,

$$\bar{\Sigma}_{t,\ell-1}^{-1} = \Sigma_\ell^{-1} + \bar{G}_{t,\ell-1}, \qquad \bar{\mu}_{t,\ell-1} = \bar{\Sigma}_{t,\ell-1}\left(\Sigma_\ell^{-1} W_\ell \psi_\ell + \bar{B}_{t,\ell-1}\right), \qquad (18)$$

and the $L$-th latent-posterior is $Q_{t,L}(\cdot) = \mathcal{N}(\bar{\mu}_{t,L}, \bar{\Sigma}_{t,L})$,

$$\bar{\Sigma}_{t,L}^{-1} = \Sigma_{L+1}^{-1} + \bar{G}_{t,L}, \qquad \bar{\mu}_{t,L} = \bar{\Sigma}_{t,L} \bar{B}_{t,L}. \qquad (19)$$

Finally, $\bar{G}_{t,\ell}$ and $\bar{B}_{t,\ell}$ for $\ell \in [L]$ are computed recursively. The basis of the recursion are

$$\bar{G}_{t,1} = W_1^\top \sum_{i=1}^K \left(\Sigma_1^{-1} - \Sigma_1^{-1} \hat{\Sigma}_{t,i} \Sigma_1^{-1}\right) W_1, \qquad \bar{B}_{t,1} = W_1^\top \Sigma_1^{-1} \sum_{i=1}^K \hat{\Sigma}_{t,i} \hat{G}_{t,i} \hat{B}_{t,i}. \qquad (20)$$

Then, the recursive step for $\ell \in [L]/\{1\}$ is,

$$\bar{G}_{t,\ell} = W_\ell^\top \left(\Sigma_\ell^{-1} - \Sigma_\ell^{-1} \bar{\Sigma}_{t,\ell-1} \Sigma_\ell^{-1}\right) W_\ell, \qquad \bar{B}_{t,\ell} = W_\ell^\top \Sigma_\ell^{-1} \bar{\Sigma}_{t,\ell-1} \bar{B}_{t,\ell-1}. \qquad (21)$$

This concludes the derivation of our posterior approximation. Note that these approximations are exact when the reward distribution follows a linear-Gaussian model, $P(\cdot \mid x; \theta_{*,a}) = \mathcal{N}(\cdot; x^\top \theta_{*,a}, \sigma^2)$.

## B.2 Derivation of Action-Posteriors for Linear Diffusion Models

To simplify derivations, we consider the case where the reward distribution is indeed linear-Gaussian as $P(\cdot \mid X_t; \theta_{*,A_t}) = \mathcal{N}\left(X_t^\top \theta_{*,A_t}, \sigma^2\right)$, but the same derivations can be applied when the rewards are non-linear. In this case, the likelihood approximation in (16) becomes exact as we have that $\mathbb{P}\left(H_{t,i} \mid \theta_{*,i} = \theta\right) \propto \mathcal{N}\left(\theta; \hat{B}_{t,i}, \hat{G}_{t,i}^{-1}\right)$, where $\hat{B}_{t,i}$ is the corresponding MLE and $\hat{G}_{t,i} = \sigma^{-2} \sum_{k \in S_{t,i}} X_k X_k^\top$ in this case. Our derivations rely on the fact that the MLE $\hat{B}_{t,i}$ in this linear-Gaussian case satisfies: $\hat{G}_{t,i} \hat{B}_{t,i} = v \sum_{k \in S_{t,i}} X_k Y_k^\top$.

**Proposition B.1.** *Consider the following model, which corresponds to the last two layers in Eq. (15)*

$$\theta_{*,i} \mid \psi_{*,1} \sim \mathcal{N}\left(W_1 \psi_{*,1}, \Sigma_1\right),$$
$$Y_t \mid X_t, \theta_{*,A_t} \sim \mathcal{N}\left(X_t^\top \theta_{*,A_t}, \sigma^2\right), \qquad\qquad \forall t \in [n].$$

*Then we have that for any $t \in [n]$ and $i \in [K]$, $P_{t,i}(\theta \mid \psi_1) = \mathbb{P}\left(\theta_{*,i} = \theta \mid \psi_{*,1} = \psi_1, H_{t,i}\right) = \mathcal{N}(\theta; \hat{\mu}_{t,i}, \hat{\Sigma}_{t,i})$, where*

$$\hat{\Sigma}_{t,i}^{-1} = \hat{G}_{t,i} + \Sigma_1^{-1}, \qquad \hat{\mu}_{t,i} = \hat{\Sigma}_{t,i}\left(\hat{G}_{t,i} \hat{B}_{t,i} + \Sigma_1^{-1} W_1 \psi_1\right).$$

*Proof.* Let $v = \sigma^{-2}$, $\quad \Lambda_1 = \Sigma_1^{-1}$. Then the action-posterior decomposes as

$$
\begin{aligned}
P_{t,i}(\theta \mid \psi_1) &= \mathbb{P}(\theta_{*,i} = \theta \mid \psi_{*,1} = \psi_1, H_{t,i}), \\
&\propto \mathbb{P}(H_{t,i} \mid \psi_{*,1} = \psi_1, \theta_{*,i} = \theta) \, \mathbb{P}(\theta_{*,i} = \theta \mid \psi_{*,1} = \psi_1), \quad \text{(Bayes rule)} \\
&= \mathbb{P}(H_{t,i} \mid \theta_{*,i} = \theta) \, \mathbb{P}(\theta_{*,i} = \theta \mid \psi_{*,1} = \psi_1), \text{ (given } \theta_{*,i}, H_{t,i} \text{ is independent of } \psi_{*,1}) \\
&= \prod_{k \in S_{t,i}} \mathcal{N}(Y_k; X_k^\top \theta, \sigma^2) \mathcal{N}(\theta; \mathrm{W}_1 \psi_1, \Sigma_1), \\
&= \exp\Big[ -\frac{1}{2}\Big(v \sum_{k \in S_{t,i}} (Y_k^2 - 2 Y_k X_k^\top \theta + (X_k^\top \theta)^2) + \theta^\top \Lambda_1 \theta - 2\theta^\top \Lambda_1 \mathrm{W}_1 \psi_1 \\
&\qquad\qquad + (\mathrm{W}_1 \psi_1)^\top \Lambda_1 (\mathrm{W}_1 \psi_1)\Big)\Big], \\
&\propto \exp\Big[ -\frac{1}{2}\Big(\theta^\top (v \sum_{k \in S_{t,i}} X_k X_k^\top + \Lambda_1)\theta - 2\theta^\top \Big(v \sum_{k \in S_{t,i}} X_k Y_k + \Lambda_1 \mathrm{W}_1 \psi_1\Big)\Big)\Big], \\
&\propto \mathcal{N}(\theta; \hat{\mu}_{t,i}, \hat{\Lambda}_{t,i}^{-1}),
\end{aligned}
$$

with $\hat{\Lambda}_{t,i} = v \sum_{k \in S_{t,i}} X_k X_k^\top + \Lambda_1$, $\hat{\Lambda}_{t,i} \hat{\mu}_{t,i} = v \sum_{k \in S_{t,i}} X_k Y_k + \Lambda_1 \mathrm{W}_1 \psi_1$. Using that, in this linear-Gaussian case, $\hat{G}_{t,i} = v \sum_{k \in S_{t,i}} X_k X_k^\top$ and $\hat{G}_{t,i} \hat{B}_{t,i} = v \sum_{k \in S_{t,i}} X_k Y_k$ concludes the proof. $\qquad\square$

The same proof applies when the reward distribution is not linear-Gaussian, with the approximation $\mathbb{P}(H_{t,i} \mid \theta_{*,i} = \theta) \approx \mathcal{N}(\theta; \hat{B}_{t,i}, \hat{G}_{t,i}^{-1})$. Using this approximation in the derivations above leads to the same results.

## B.3 Derivation of recursive latent-posteriors for linear diffusion models

Again, to simplify derivations, we consider the case where the reward distribution is indeed linear-Gaussian as $P(\cdot \mid X_t; \theta_{*,A_t}) = \mathcal{N}(X_t^\top \theta_{*,A_t}, \sigma^2)$, but the same derivations can be applied when the rewards are non-linear.

**Proposition B.2.** *For any $\ell \in [L]/\{1\}$, the $\ell - 1$-th conditional latent-posterior reads $Q_{t,\ell-1}(\cdot \mid \psi_\ell) = \mathcal{N}(\bar{\mu}_{t,\ell-1}, \bar{\Sigma}_{t,\ell-1})$, with*

$$
\bar{\Sigma}_{t,\ell-1}^{-1} = \Sigma_\ell^{-1} + \bar{G}_{t,\ell-1}, \qquad\qquad \bar{\mu}_{t,\ell-1} = \bar{\Sigma}_{t,\ell-1}\big(\Sigma_\ell^{-1} \mathrm{W}_\ell \psi_\ell + \bar{B}_{t,\ell-1}\big), \tag{22}
$$

*and the $L$-th latent-posterior reads $Q_{t,L}(\cdot) = \mathcal{N}(\bar{\mu}_{t,L}, \bar{\Sigma}_{t,L})$, with*

$$
\bar{\Sigma}_{t,L}^{-1} = \Sigma_{L+1}^{-1} + \bar{G}_{t,L}, \qquad\qquad \bar{\mu}_{t,L} = \bar{\Sigma}_{t,L} \bar{B}_{t,L}. \tag{23}
$$

*Proof.* Let $\ell \in [L]/\{1\}$. Then, Bayes rule yields that

$$
Q_{t,\ell-1}(\psi_{\ell-1} \mid \psi_\ell) \propto \mathbb{P}(H_t \mid \psi_{*,\ell-1} = \psi_{\ell-1}) \mathcal{N}(\psi_{\ell-1}, \mathrm{W}_\ell \psi_\ell, \Sigma_\ell),
$$

But from [Lemma B.3](), we know that

$$
\mathbb{P}(H_t \mid \psi_{*,\ell-1} = \psi_{\ell-1}) \propto \exp\Big[-\frac{1}{2}\psi_{\ell-1}^\top \bar{G}_{t,\ell-1} \psi_{\ell-1} + \psi_{\ell-1}^\top \bar{B}_{t,\ell-1}\Big].
$$

Therefore,

$$
\begin{aligned}
Q_{t,\ell-1}(\psi_{\ell-1} \mid \psi_\ell) &\propto \exp\Big[-\frac{1}{2}\psi_{\ell-1}^\top \bar{G}_{t,\ell-1} \psi_{\ell-1} + \psi_{\ell-1}^\top \bar{B}_{t,\ell-1}\Big]\mathcal{N}(\psi_{\ell-1}, \mathrm{W}_\ell \psi_\ell, \Sigma_\ell), \\
&\propto \exp\Big[-\frac{1}{2}\psi_{\ell-1}^\top \bar{G}_{t,\ell-1} \psi_{\ell-1} + \psi_{\ell-1}^\top \bar{B}_{t,\ell-1} \\
&\qquad\qquad - \frac{1}{2}(\psi_{\ell-1} - \mathrm{W}_\ell \psi_\ell)^\top \Sigma_\ell^{-1}(\psi_{\ell-1} - \mathrm{W}_\ell \psi_\ell))\Big], \\
&\overset{(i)}{\propto} \exp\Big[-\frac{1}{2}\psi_{\ell-1}^\top (\bar{G}_{t,\ell-1} + \Sigma_\ell^{-1})\psi_{\ell-1} + \psi_{\ell-1}^\top (\bar{B}_{t,\ell-1} + \Sigma_\ell^{-1}\mathrm{W}_\ell \psi_\ell)\Big], \\
&\overset{(ii)}{\propto} \mathcal{N}(\psi_{\ell-1}; \bar{\mu}_{t,\ell-1}, \bar{\Sigma}_{t,\ell-1}),
\end{aligned}
$$

with $\bar{\Sigma}_{t,\ell-1}^{-1} = \Sigma_\ell^{-1} + \bar{G}_{t,\ell-1}$ and $\bar{\mu}_{t,\ell-1} = \bar{\Sigma}_{t,\ell-1}\big(\Sigma_\ell^{-1}\mathrm{W}_\ell\psi_\ell + \bar{B}_{t,\ell-1}\big)$. In $(i)$, we omit terms that are constant in $\psi_{\ell-1}$. In $(ii)$, we complete the square. This concludes the proof for $\ell \in [L]/\{1\}$. For $Q_{t,L}$, we use Bayes rule to get

$$Q_{t,L}(\psi_L) \propto \mathbb{P}\left(H_t \mid \psi_{*,L} = \psi_L\right)\mathcal{N}(\psi_L, 0, \Sigma_{L+1}).$$

Then from Lemma B.3, we know that

$$\mathbb{P}\left(H_t \mid \psi_{*,L} = \psi_L\right) \propto \exp\left[-\frac{1}{2}\psi_L^\top \bar{G}_{t,L}\psi_L + \psi_L^\top \bar{B}_{t,L}\right],$$

We then use the same derivations above to compute the product $\exp\left[-\frac{1}{2}\psi_L^\top \bar{G}_{t,L}\psi_L + \psi_L^\top \bar{B}_{t,L}\right] \times \mathcal{N}(\psi_L, 0, \Sigma_{L+1})$, which concludes the proof. $\qquad\square$

**Lemma B.3.** *The following holds for any $t \in [n]$ and $\ell \in [L]$,*

$$\mathbb{P}\left(H_t \mid \psi_{*,\ell} = \psi_\ell\right) \propto \exp\left[-\frac{1}{2}\psi_\ell^\top \bar{G}_{t,\ell}\psi_\ell + \psi_\ell^\top \bar{B}_{t,\ell}\right],$$

*where $\bar{G}_{t,\ell}$ and $\bar{B}_{t,\ell}$ are defined by recursion in Section 3.1.*

*Proof.* We prove this result by induction. To reduce clutter, we let $v = \sigma^{-2}$, and $\Lambda_1 = \Sigma_1^{-1}$. We start with the base case of the induction when $\ell = 1$.

**(I) Base case.** Here we want to show that $\mathbb{P}\left(H_t \mid \psi_{*,1} = \psi_1\right) \propto \exp\left[-\frac{1}{2}\psi_1^\top \bar{G}_{t,1}\psi_1 + \psi_1^\top \bar{B}_{t,1}\right]$, where $\bar{G}_{t,1}$ and $\bar{B}_{t,1}$ are given in Eq. (20). First, we have that

$$\mathbb{P}\left(H_t \mid \psi_{*,1} = \psi_1\right) \overset{(i)}{=} \prod_{i\in[K]} \mathbb{P}\left(H_{t,i} \mid \psi_{*,1} = \psi_1\right) = \prod_{i\in[K]} \int_\theta \mathbb{P}\left(H_{t,i}, \theta_{*,i} = \theta \mid \psi_{*,1} = \psi_1\right)\mathrm{d}\theta,$$

$$= \prod_{i\in[K]} \int_\theta \mathbb{P}\left(H_{t,i} \mid \theta_{*,i} = \theta\right)\mathcal{N}\left(\theta; \mathrm{W}_1\psi_1, \Sigma_1\right)\mathrm{d}\theta,$$

$$= \prod_{i\in[K]} \underbrace{\int_\theta \Big(\prod_{k\in S_{t,i}} \mathcal{N}(Y_k; X_k^\top\theta, \sigma^2)\Big)\mathcal{N}\left(\theta; \mathrm{W}_1\psi_1, \Sigma_1\right)\mathrm{d}\theta}_{h_i(\psi_1)},$$

$$= \prod_{i\in[K]} h_i(\psi_1), \tag{24}$$

where $(i)$ follows from the fact that $\theta_{*,i}$ for $i \in [K]$ are conditionally independent given $\psi_{*,1} = \psi_1$ and that given $\theta_{*,i}$, $H_{t,i}$ is independent of $\psi_{*,1}$. Now we compute $h_i(\psi_1) = \int_\theta \Big(\prod_{k\in S_{t,i}} \mathcal{N}(Y_k; X_k^\top\theta, \sigma^2)\Big)\mathcal{N}\left(\theta; \mathrm{W}_1\psi_1, \Sigma_1\right)\mathrm{d}\theta$ as

$$h_i(\psi_1) = \int_\theta \Big(\prod_{k\in S_{t,i}} \mathcal{N}(Y_k; X_k^\top\theta, \sigma^2)\Big)\mathcal{N}(\theta; \mathrm{W}_1\psi_1, \Sigma_1)\,\mathrm{d}\theta,$$

$$\propto \int_\theta \exp\left[-\frac{1}{2}v \sum_{k\in S_{t,i}} (Y_k - X_k^\top\theta)^2 - \frac{1}{2}(\theta - \mathrm{W}_1\psi_1)^\top\Lambda_1(\theta - \mathrm{W}_1\psi_1)\right]\mathrm{d}\theta,$$

$$= \int_\theta \exp\Big[-\frac{1}{2}\Big(v \sum_{k\in S_{t,i}} (Y_k^2 - 2Y_k\theta^\top X_k + (\theta^\top X_k)^2) + \theta^\top\Lambda_1\theta - 2\theta^\top\Lambda_1\mathrm{W}_1\psi_1$$

$$+ (\mathrm{W}_1\psi_1)^\top\Lambda_1(\mathrm{W}_1\psi_1)\Big)\Big]\mathrm{d}\theta,$$

$$\propto \int_\theta \exp\Big[-\frac{1}{2}\Big(\theta^\top\Big(v \sum_{k\in S_{t,i}} X_k X_k^\top + \Lambda_1\Big)\theta - 2\theta^\top\Big(v \sum_{k\in S_{t,i}} Y_k X_k$$

$$+ \Lambda_1\mathrm{W}_1\psi_1\Big) + (\mathrm{W}_1\psi_1)^\top\Lambda_1(\mathrm{W}_1\psi_1)\Big)\Big]\mathrm{d}\theta.$$

But we know that $\hat{G}_{t,i} = v \sum_{k \in S_{t,i}} X_k X_k^\top$, and $\hat{G}_{t,i} \hat{B}_{t,i} = v \sum_{k \in S_{t,i}} Y_k X_k$ (because we assumed linear-Gaussian likelihood). To further simplify expressions, we also let

$$V = \left(\hat{G}_{t,i} + \Lambda_1\right)^{-1}, \quad U = V^{-1}, \quad \beta = V\left(\hat{G}_{t,i}\hat{B}_{t,i} + \Lambda_1 W_1 \psi_1\right).$$

We have that $UV = VU = I_d$, and thus

$$h_i(\psi_1) \propto \int_\theta \exp\left[-\frac{1}{2}\left(\theta^\top U \theta - 2\theta^\top UV\left(\hat{G}_{t,i}\hat{B}_{t,i} + \Lambda_1 W_1 \psi_1\right) + (W_1\psi_1)^\top \Lambda_1 (W_1\psi_1)\right)\right] d\theta,$$

$$= \int_\theta \exp\left[-\frac{1}{2}\left(\theta^\top U \theta - 2\theta^\top U \beta + (W_1\psi_1)^\top \Lambda_1 (W_1\psi_1)\right)\right] d\theta,$$

$$= \int_\theta \exp\left[-\frac{1}{2}\left((\theta - \beta)^\top U(\theta - \beta) - \beta^\top U\beta + (W_1\psi_1)^\top \Lambda_1 (W_1\psi_1)\right)\right] d\theta,$$

$$\propto \exp\left[-\frac{1}{2}\left(-\beta^\top U\beta + (W_1\psi_1)^\top \Lambda_1 (W_1\psi_1)\right)\right],$$

$$= \exp\left[-\frac{1}{2}\left(-\left(\hat{G}_{t,i}\hat{B}_{t,i} + \Lambda_1 W_1 \psi_1\right)^\top V\left(\hat{G}_{t,i}\hat{B}_{t,i} + \Lambda_1 W_1 \psi_1\right) + (W_1\psi_1)^\top \Lambda_1 (W_1\psi_1)\right)\right],$$

$$\propto \exp\left[-\frac{1}{2}\left(\psi_1^\top W_1^\top (\Lambda_1 - \Lambda_1 V \Lambda_1) W_1 \psi_1 - 2\psi_1^\top \left(W_1^\top \Lambda_1 V \hat{G}_{t,i}\hat{B}_{t,i}\right)\right)\right],$$

$$= \exp\left[-\frac{1}{2}\psi_1^\top \Omega_i \psi_1 + \psi_1^\top m_i\right],$$

where

$$\Omega_i = W_1^\top (\Lambda_1 - \Lambda_1 V \Lambda_1) W_1 = W_1^\top \left(\Lambda_1 - \Lambda_1(\hat{G}_{t,i} + \Lambda_1)^{-1}\Lambda_1\right) W_1,$$
$$m_i = W_1^\top \Lambda_1 V \hat{G}_{t,i}\hat{B}_{t,i} = W_1^\top \Lambda_1(\hat{G}_{t,i} + \Lambda_1)^{-1}\hat{G}_{t,i}\hat{B}_{t,i}. \tag{25}$$

But notice that $V = (\hat{G}_{t,i} + \Lambda_1)^{-1} = \hat{\Sigma}_{t,i}$ and thus

$$\Omega_i = W_1^\top \left(\Lambda_1 - \Lambda_1 \hat{\Sigma}_{t,i} \Lambda_1\right) W_1, \qquad\qquad m_i = W_1^\top \Lambda_1 \hat{\Sigma}_{t,i} \hat{G}_{t,i}\hat{B}_{t,i}. \tag{26}$$

Finally, we plug this result in Eq. (24) to get

$$\mathbb{P}\left(H_t \mid \psi_{*,1} = \psi_1\right) = \prod_{i \in [K]} h_i(\psi_1) \propto \prod_{i \in [K]} \exp\left[-\frac{1}{2}\psi_1^\top \Omega_i \psi_1 + \psi_1^\top m_i\right],$$

$$= \exp\left[-\frac{1}{2}\psi_1^\top \sum_{i \in [K]} \Omega_i \psi_1 + \psi_1^\top \sum_{i \in [K]} m_i\right],$$

$$= \exp\left[-\frac{1}{2}\psi_1^\top \bar{G}_{t,1}\psi_1 + \psi_1^\top \bar{B}_{t,1}\right],$$

where

$$\bar{G}_{t,1} = \sum_{i=1}^K \Omega_i = \sum_{i=1}^K W_1^\top \left(\Lambda_1 - \Lambda_1 \hat{\Sigma}_{t,i} \Lambda_1\right) W_1 = W_1^\top \sum_{i=1}^K \left(\Sigma_1^{-1} - \Sigma_1^{-1}\hat{\Sigma}_{t,i}\Sigma_1^{-1}\right) W_1,$$

$$\bar{B}_{t,1} = \sum_{i=1}^K m_i = \sum_{i=1}^K \hat{\Sigma}_{t,i}\hat{G}_{t,i}\hat{B}_{t,i} = W_1^\top \Sigma_1^{-1} \sum_{i=1}^K \hat{\Sigma}_{t,i}\hat{G}_{t,i}\hat{B}_{t,i}.$$

This concludes the proof of the base case.

**(II) Induction step.** Let $\ell \in [L]/\{1\}$. Suppose that

$$\mathbb{P}\left(H_t \mid \psi_{*,\ell-1} = \psi_{\ell-1}\right) \propto \exp\left[-\frac{1}{2}\psi_{\ell-1}^\top \bar{G}_{t,\ell-1}\psi_{\ell-1} + \psi_{\ell-1}^\top \bar{B}_{t,\ell-1}\right]. \tag{27}$$

Then we want to show that

$$\mathbb{P}\left(H_t \mid \psi_{*,\ell} = \psi_\ell\right) \propto \exp\left[-\frac{1}{2}\psi_\ell^\top \bar{G}_{t,\ell}\psi_\ell + \psi_\ell^\top \bar{B}_{t,\ell}\right],$$

where

$$\bar{G}_{t,\ell} = W_\ell^\top \left(\Sigma_\ell^{-1} - \Sigma_\ell^{-1}\bar{\Sigma}_{t,\ell-1}\Sigma_\ell^{-1}\right)W_\ell = W_\ell^\top \left(\Sigma_\ell^{-1} - \Sigma_\ell^{-1}(\Sigma_\ell^{-1} + \bar{G}_{t,\ell-1})^{-1}\Sigma_\ell^{-1}\right)W_\ell,$$

$$\bar{B}_{t,\ell} = W_\ell^\top \Sigma_\ell^{-1}\bar{\Sigma}_{t,\ell-1}\bar{B}_{t,\ell-1} = W_\ell^\top \Sigma_\ell^{-1}(\Sigma_\ell^{-1} + \bar{G}_{t,\ell-1})^{-1}\bar{B}_{t,\ell-1}.$$

To achieve this, we start by expressing $\mathbb{P}\left(H_t \mid \psi_{*,\ell} = \psi_\ell\right)$ in terms of $\mathbb{P}\left(H_t \mid \psi_{*,\ell-1} = \psi_{\ell-1}\right)$ as

$$\mathbb{P}\left(H_t \mid \psi_{*,\ell} = \psi_\ell\right) = \int_{\psi_{\ell-1}} \mathbb{P}\left(H_t, \psi_{*,\ell-1} = \psi_{\ell-1} \mid \psi_{*,\ell} = \psi_\ell\right) \mathrm{d}\psi_{\ell-1},$$

$$= \int_{\psi_{\ell-1}} \mathbb{P}\left(H_t \mid \psi_{*,\ell-1} = \psi_{\ell-1}, \psi_{*,\ell} = \psi_\ell\right) \mathcal{N}(\psi_{\ell-1}; W_\ell\psi_\ell, \Sigma_\ell) \, \mathrm{d}\psi_{\ell-1},$$

$$= \int_{\psi_{\ell-1}} \mathbb{P}\left(H_t \mid \psi_{*,\ell-1} = \psi_{\ell-1}\right) \mathcal{N}(\psi_{\ell-1}; W_\ell\psi_\ell, \Sigma_\ell) \, \mathrm{d}\psi_{\ell-1},$$

$$\propto \int_{\psi_{\ell-1}} \exp\left[-\frac{1}{2}\psi_{\ell-1}^\top \bar{G}_{t,\ell-1}\psi_{\ell-1} + \psi_{\ell-1}^\top \bar{B}_{t,\ell-1}\right] \mathcal{N}(\psi_{\ell-1}; W_\ell\psi_\ell, \Sigma_\ell) \, \mathrm{d}\psi_{\ell-1},$$

$$\propto \int_{\psi_{\ell-1}} \exp\left[-\frac{1}{2}\psi_{\ell-1}^\top \bar{G}_{t,\ell-1}\psi_{\ell-1} + \psi_{\ell-1}^\top \bar{B}_{t,\ell-1}\right.$$
$$\left. + (\psi_{\ell-1} - W_\ell\psi_\ell)^\top \Lambda_\ell(\psi_{\ell-1} - W_\ell\psi_\ell)\right) \mathrm{d}\psi_{\ell-1}.$$

Now let $S = \bar{G}_{t,\ell-1} + \Lambda_\ell$ and $V = \bar{B}_{t,\ell-1} + \Lambda_\ell W_\ell\psi_\ell$. Then we have that,

$$\mathbb{P}\left(H_t \mid \psi_{*,\ell} = \psi_\ell\right)$$

$$\propto \int_{\psi_{\ell-1}} \exp\left[-\frac{1}{2}\psi_{\ell-1}^\top \bar{G}_{t,\ell-1}\psi_{\ell-1} + \psi_{\ell-1}^\top \bar{B}_{t,\ell-1}\right.$$
$$\left. + (\psi_{\ell-1} - W_\ell\psi_\ell)^\top \Lambda_\ell(\psi_{\ell-1} - W_\ell\psi_\ell)\right)\right] \mathrm{d}\psi_{\ell-1},$$

$$\propto \int_{\psi_{\ell-1}} \exp\left[-\frac{1}{2}\left(\psi_{\ell-1}^\top S\psi_{\ell-1} - 2\psi_{\ell-1}^\top \left(\bar{B}_{t,\ell-1} + \Lambda_\ell W_\ell\psi_\ell\right) + \psi_\ell^\top W_\ell^\top \Lambda_\ell W_\ell\psi_\ell\right)\right] \mathrm{d}\psi_{\ell-1},$$

$$= \int_{\psi_{\ell-1}} \exp\left[-\frac{1}{2}\left(\psi_{\ell-1}^\top S(\psi_{\ell-1} - 2S^{-1}V) + \psi_\ell^\top W_\ell^\top \Lambda_\ell W_\ell\psi_\ell\right)\right] \mathrm{d}\psi_{\ell-1},$$

$$= \int_{\psi_{\ell-1}} \exp\left[-\frac{1}{2}\left((\psi_{\ell-1} - S^{-1}V)^\top S(\psi_{\ell-1} - S^{-1}V)\right.\right.$$
$$\left.\left. + \psi_\ell^\top W_\ell^\top \Lambda_\ell W_\ell\psi_\ell - V^\top S^{-1}V\right)\right] \mathrm{d}\psi_{\ell-1}.$$

In the second step, we omit constants in $\psi_\ell$ and $\psi_{\ell-1}$. Thus

$$\mathbb{P}\left(H_t \mid \psi_{*,\ell} = \psi_\ell\right)$$

$$\propto \int_{\psi_{\ell-1}} \exp\left[-\frac{1}{2}\left((\psi_{\ell-1} - S^{-1}V)^\top S(\psi_{\ell-1} - S^{-1}V) + \psi_\ell^\top W_\ell^\top \Lambda_\ell W_\ell\psi_\ell - V^\top S^{-1}V\right)\right] \mathrm{d}\psi_{\ell-1},$$

$$\propto \exp\left[-\frac{1}{2}\left(\psi_\ell^\top W_\ell^\top \Lambda_\ell W_\ell\psi_\ell - V^\top S^{-1}V\right)\right].$$

It follows that

$$
\begin{aligned}
&\mathbb{P}\left(H_t \,|\, \psi_{*,\ell} = \psi_\ell\right) \\
&\propto \exp\left[-\frac{1}{2}\left(\psi_\ell^\top \mathrm{W}_\ell^\top \Lambda_\ell \mathrm{W}_\ell \psi_\ell - V^\top S^{-1} V\right)\right], \\
&= \exp\left[-\frac{1}{2}\left(\psi_\ell^\top \mathrm{W}_\ell^\top \Lambda_\ell \mathrm{W}_\ell \psi_\ell - \left(\bar{B}_{t,\ell-1} + \Lambda_\ell \mathrm{W}_\ell \psi_\ell\right)^\top S^{-1}\left(\bar{B}_{t,\ell-1} + \Lambda_\ell \mathrm{W}_\ell \psi_\ell\right)\right)\right] \\
&\propto \exp\left[-\frac{1}{2}\left(\psi_\ell^\top \left(\mathrm{W}_\ell^\top \Lambda_\ell \mathrm{W}_\ell - \mathrm{W}_\ell^\top \Lambda_\ell S^{-1} \Lambda_\ell \mathrm{W}_\ell\right)\psi_\ell - 2\psi_\ell^\top \mathrm{W}_\ell^\top \Lambda_\ell S^{-1} \bar{B}_{t,\ell-1}\right)\right], \\
&= \exp\left[-\frac{1}{2}\psi_\ell^\top \bar{G}_{t,\ell}\psi_\ell + \psi_\ell^\top \bar{B}_{t,\ell}\right].
\end{aligned}
$$

In the last step, we omit constants in $\psi_\ell$ and we set

$$
\begin{aligned}
\bar{G}_{t,\ell} &= \mathrm{W}_\ell^\top \left(\Lambda_\ell - \Lambda_\ell S^{-1} \Lambda_\ell\right)\mathrm{W}_\ell = \mathrm{W}_\ell^\top \left(\Lambda_\ell - \Lambda_\ell(\Lambda_\ell + \bar{G}_{t,\ell-1})^{-1}\Sigma_\ell^{-1}\Lambda_\ell\right)\mathrm{W}_\ell\,, \\
\bar{B}_{t,\ell} &= \mathrm{W}_\ell^\top \Lambda_\ell S^{-1} \bar{B}_{t,\ell-1} = \mathrm{W}_\ell^\top \Lambda_\ell(\Lambda_\ell + \bar{G}_{t,\ell-1})^{-1}\bar{B}_{t,\ell-1}\,.
\end{aligned}
$$

This completes the proof. $\qquad\square$

Similarly, this same proof applies when the reward distribution is not linear-Gaussian, with the approximation $\mathbb{P}\left(H_{t,i} \,|\, \theta_{*,i} = \theta\right) \approx \mathcal{N}\left(\theta; \hat{B}_{t,i}, \hat{G}_{t,i}^{-1}\right)$. Using this approximation in the derivations above leads to the same results.

# C   Posterior derivations for non-linear diffusion models

After deriving the posteriors for linear score functions $f_\ell$, we now get back to the general case in (1), where the score functions are potentially non-linear. Approximation is needed since both the score functions and rewards can be non-linear. To avoid any computational challenges, we use a simple and intuitive approximation, where all posteriors $P_{t,i}$ and $Q_{t,\ell}$ are approximated by the Gaussian distributions in Appendix B.1, with few changes. First, the terms $\mathrm{W}_\ell \psi_\ell$ in (18) are replaced by $f_\ell(\psi_\ell)$. This accounts for the fact that the prior mean is now $f_\ell(\psi_\ell)$ rather than $\mathrm{W}_\ell \psi_\ell$, and this is the main difference between the linear diffusion model in (15) and the general, potentially non-linear, diffusion model in (1). Second, the matrix multiplications that involve the matrices $\mathrm{W}_\ell$ in (20) and (21) are simply removed. Despite being simple, this approximation is efficient and avoids the computational burden of heavy approximate sampling algorithms required for each latent parameter. This is why deriving the exact posterior for linear score functions was key beyond enabling theoretical analyses. Moreover, this approximation retains some key attributes of exact posteriors. Specifically, in the absence of data, it recovers precisely the prior in (1), and as more data is accumulated, the influence of the prior diminishes.

# D   Regret proof and additional discussions

## D.1   Sketch of the proof

We start with the following standard lemma upon which we build our analysis [Aouali et al., 2023b].

**Lemma D.1.** *Assume that* $\mathbb{P}\left(\theta_{*,i} = \theta \,|\, H_t\right) = \mathcal{N}(\theta; \breve{\mu}_{t,i}, \breve{\Sigma}_{t,i})$ *for any* $i \in [K]$, *then for any* $\delta \in (0,1)$,

$$
\mathcal{BR}(n) \le \sqrt{2n \log(1/\delta)}\sqrt{\mathbb{E}\left[\sum_{t=1}^{n} \|X_t\|_{\breve{\Sigma}_{t,A_t}}^2\right]} + cn\delta\,, \qquad \text{where } c > 0 \text{ is a constant}. \tag{28}
$$

Applying Lemma D.1 requires proving that the *marginal* action-posteriors $\mathbb{P}\left(\theta_{*,i} = \theta \,|\, H_t\right)$ in Eq. (3) are Gaussian and computing their covariances, while we only know the *conditional* action-posteriors $P_{t,i}$ and latent-posteriors $Q_{t,\ell}$. This is achieved by leveraging the preservation properties of the family of Gaussian distributions [Koller and Friedman, 2009] and the total covariance decomposition [Weiss, 2005] which leads to the next lemma.

**Lemma D.2.** *Let $t \in [n]$ and $i \in [K]$, then the marginal covariance matrix $\check{\Sigma}_{t,i}$ reads*

$$\check{\Sigma}_{t,i} = \hat{\Sigma}_{t,i} + \sum_{\ell \in [L]} \mathrm{P}_{i,\ell} \bar{\Sigma}_{t,\ell} \mathrm{P}_{i,\ell}^\top, \quad \text{where } \mathrm{P}_{i,\ell} = \hat{\Sigma}_{t,i} \Sigma_1^{-1} \mathrm{W}_1 \prod_{k=1}^{\ell-1} \bar{\Sigma}_{t,k} \Sigma_{k+1}^{-1} \mathrm{W}_{k+1}. \quad (29)$$

The marginal covariance matrix $\check{\Sigma}_{t,i}$ in Eq. (29) decomposes into $L + 1$ terms. The first term corresponds to the posterior uncertainty of $\theta_{*,i} \mid \psi_{*,1}$. The remaining $L$ terms capture the posterior uncertainties of $\psi_{*,L}$ and $\psi_{*,\ell-1} \mid \psi_{*,\ell}$ for $\ell \in [L]/\{1\}$. These are then used to quantify the posterior information gain of latent parameters after one round as follows.

**Lemma D.3** (Posterior information gain). *Let $t \in [n]$ and $\ell \in [L]$, then*

$$\bar{\Sigma}_{t+1,\ell}^{-1} - \bar{\Sigma}_{t,\ell}^{-1} \succeq \sigma^{-2} \sigma_{\mathrm{MAX}}^{-2\ell} \mathrm{P}_{A_t,\ell}^\top X_t X_t^\top \mathrm{P}_{A_t,\ell}, \quad \text{where } \sigma_{\mathrm{MAX}}^2 = \max_{\ell \in [L+1]} 1 + \frac{\sigma_\ell^2}{\sigma^2}. \quad (30)$$

Finally, Lemma D.2 is used to decompose $\|X_t\|_{\check{\Sigma}_{t,A_t}}^2$ in Eq. (28) into $L + 1$ terms. Each term is bounded thanks to Lemma D.3. This results in the Bayes regret bound in Theorem 4.1.

## D.2 Technical contributions

Our main technical contributions are the following.

**Lemma D.2.** In dTS, sampling is done hierarchically, meaning the marginal posterior distribution of $\theta_{*,i}|H_t$ is not explicitly defined. Instead, we use the conditional posterior distribution of $\theta_{*,i}|H_t, \psi_{*,1}$. The first contribution was deriving $\theta_{*,i}|H_t$ using the total covariance decomposition combined with an induction proof, as our posteriors in Section 3.1 were derived recursively. Unlike in Bayes regret analysis for standard Thompson sampling, where the posterior distribution of $\theta_{*,i}|H_t$ is predetermined due to the absence of latent parameters, our method necessitates this recursive total covariance decomposition, marking a first difference from the standard Bayesian proofs of Thompson sampling. Note that HierTS, which is developed for multi-task linear bandits, also employs total covariance decomposition, but it does so under the assumption of a single latent parameter; on which action parameters are centered. Our extension significantly differs as it is tailored for contextual bandits with multiple, successive levels of latent parameters, moving away from HierTS's assumption of a 1-level structure. Roughly speaking, HierTS when applied to contextual would consider a single-level hierarchy, where $\theta_{*,i}|\psi_{*,1} \sim \mathcal{N}(\psi_{*,1}, \Sigma_1)$ with $L = 1$. In contrast, our model proposes a multi-level hierarchy, where the first level is $\theta_{*,i}|\psi_{*,1} \sim \mathcal{N}(W_1\psi_{*,1}, \Sigma_1)$. This also introduces a new aspect to our approach – the use of a linear function $W_1\psi_{*,1}$, as opposed to HierTS's assumption where action parameters are centered directly on the latent parameter. Thus, while HierTS also uses the total covariance decomposition, our generalize it to multi-level hierarchies under $L$ linear functions $W_\ell\psi_{*,\ell}$, instead of a single-level hierarchy under a single identity function $\psi_{*,1}$.

**Lemma D.3.** In Bayes regret proofs for standard Thompson sampling, we often quantify the posterior information gain. This is achieved by monitoring the increase in posterior precision for the action taken $A_t$ in each round $t \in [n]$. However, in dTS, our analysis extends beyond this. We not only quantify the posterior information gain for the taken action but also for every latent parameter, since they are also learned. This lemma addresses this aspect. To elaborate, we use the recursive formulas in Section 3.1 that connect the posterior covariance of each latent parameter $\psi_{*,\ell}$ with the covariance of the posterior action parameters $\theta_{*,i}$. This allows us to propagate the information gain associated with the action taken in round $A_t$ to all latent parameters $\psi_{*,\ell}$, for $\ell \in [L]$ by induction. This is a novel contribution, as it is not a feature of Bayes regret analyses in standard Thompson sampling.

**Proposition 4.2.** Building upon the insights of Theorem 4.1, we introduce the sparsity assumption **(A3)**. Under this assumption, we demonstrate that the Bayes regret outlined in Theorem 4.1 can be significantly refined. Specifically, the regret becomes contingent on dimensions $d_\ell \leq d$, as opposed to relying on the entire dimension $d$. This sparsity assumption is both a novel and a key technical contribution to our work. Its underlying principle is straightforward: the Bayes regret is influenced by the quantity of parameters that require learning. With the sparsity assumption, this number is reduced to less than $d$ for each latent parameter. To substantiate this claim, we revisit the proof of Theorem 4.1 and modify a crucial equality. This adjustment results in a more precise representation by partitioning the covariance matrix of each latent parameter $\psi_{*,\ell}$ into blocks. These blocks comprise a $d_\ell \times d_\ell$ segment corresponding to the learnable $d_\ell$ parameters of $\psi_{*,\ell}$, and another block of size $(d - d_\ell) \times (d - d_\ell)$ that does not necessitate learning. This decomposition allows us to conclude that the final regret is solely dependent on $d_\ell$, marking a significant refinement from the original theorem.

### D.3 Proof of lemma D.2

In this proof, we heavily rely on the total covariance decomposition [Weiss, 2005]. Also, refer to [Hong et al., 2022b, Section 5.2] for a brief introduction to this decomposition. Now, from Eq. (17), we have that

$$\text{cov}\left[\theta_{*,i} \,|\, H_t, \psi_{*,1}\right] = \hat{\Sigma}_{t,i} = \left(\hat{G}_{t,i} + \Sigma_1^{-1}\right)^{-1},$$

$$\mathbb{E}\left[\theta_{*,i} \,|\, H_t, \psi_{*,1}\right] = \hat{\mu}_{t,i} = \hat{\Sigma}_{t,i}\left(\hat{G}_{t,i}\hat{B}_{t,i} + \Sigma_1^{-1}W_1\psi_{*,1}\right).$$

First, given $H_t$, $\text{cov}\left[\theta_{*,i} \,|\, H_t, \psi_{*,1}\right] = \left(\hat{G}_{t,i} + \Sigma_1^{-1}\right)^{-1}$ is constant. Thus

$$\mathbb{E}\left[\text{cov}\left[\theta_{*,i} \,|\, H_t, \psi_{*,1}\right] | H_t\right] = \text{cov}\left[\theta_{*,i} \,|\, H_t, \psi_{*,1}\right] = \left(\hat{G}_{t,i} + \Sigma_1^{-1}\right)^{-1} = \hat{\Sigma}_{t,i}.$$

In addition, given $H_t$, $\hat{\Sigma}_{t,i}$, $\hat{G}_{t,i}$ and $\hat{B}_{t,i}$ are constant. Thus

$$\begin{aligned}
\text{cov}\left[\mathbb{E}\left[\theta_{*,i} \,|\, H_t, \psi_{*,1}\right] | H_t\right] &= \text{cov}\left[\hat{\Sigma}_{t,i}\left(\hat{G}_{t,i}\hat{B}_{t,i} + \Sigma_1^{-1}W_1\psi_{*,1}\right) \Big| H_t\right], \\
&= \text{cov}\left[\hat{\Sigma}_{t,i}\Sigma_1^{-1}W_1\psi_{*,1} \Big| H_t\right], \\
&= \hat{\Sigma}_{t,i}\Sigma_1^{-1}W_1\text{cov}\left[\psi_{*,1} \,|\, H_t\right]W_1^\top\Sigma_1^{-1}\hat{\Sigma}_{t,i}, \\
&= \hat{\Sigma}_{t,i}\Sigma_1^{-1}W_1\bar{\bar{\Sigma}}_{t,1}W_1^\top\Sigma_1^{-1}\hat{\Sigma}_{t,i},
\end{aligned}$$

where $\bar{\bar{\Sigma}}_{t,1} = \text{cov}\left[\psi_{*,1} \,|\, H_t\right]$ is the marginal posterior covariance of $\psi_{*,1}$. Finally, the total covariance decomposition [Weiss, 2005, Hong et al., 2022b] yields that

$$\begin{aligned}
\check{\Sigma}_{t,i} = \text{cov}\left[\theta_{*,i} \,|\, H_t\right] &= \mathbb{E}\left[\text{cov}\left[\theta_{*,i} \,|\, H_t, \psi_{*,1}\right] | H_t\right] + \text{cov}\left[\mathbb{E}\left[\theta_{*,i} \,|\, H_t, \psi_{*,1}\right] | H_t\right], \\
&= \hat{\Sigma}_{t,i} + \hat{\Sigma}_{t,i}\Sigma_1^{-1}W_1\bar{\bar{\Sigma}}_{t,1}W_1^\top\Sigma_1^{-1}\hat{\Sigma}_{t,i},
\end{aligned} \tag{31}$$

However, $\bar{\bar{\Sigma}}_{t,1} = \text{cov}\left[\psi_{*,1} \,|\, H_t\right]$ is different from $\bar{\Sigma}_{t,1} = \text{cov}\left[\psi_{*,1} \,|\, H_t, \psi_{*,2}\right]$ that we already derived in Eq. (18). Thus we do not know the expression of $\bar{\bar{\Sigma}}_{t,1}$. But we can use the same total covariance decomposition trick to find it. Precisely, let $\bar{\bar{\Sigma}}_{t,\ell} = \text{cov}\left[\psi_{*,\ell} \,|\, H_t\right]$ for any $\ell \in [L]$. Then we have that

$$\bar{\Sigma}_{t,1} = \text{cov}\left[\psi_{*,1} \,|\, H_t, \psi_{*,2}\right] = \left(\Sigma_2^{-1} + \bar{G}_{t,1}\right)^{-1},$$

$$\bar{\mu}_{t,1} = \mathbb{E}\left[\psi_{*,1} \,|\, H_t, \psi_{*,2}\right] = \bar{\Sigma}_{t,1}\left(\Sigma_2^{-1}W_2\psi_{*,2} + \bar{B}_{t,1}\right).$$

First, given $H_t$, $\text{cov}\left[\psi_{*,1} \,|\, H_t, \psi_{*,2}\right] = \left(\Sigma_2^{-1} + \bar{G}_{t,1}\right)^{-1}$ is constant. Thus

$$\mathbb{E}\left[\text{cov}\left[\psi_{*,1} \,|\, H_t, \psi_{*,2}\right] | H_t\right] = \text{cov}\left[\psi_{*,1} \,|\, H_t, \psi_{*,2}\right] = \bar{\Sigma}_{t,1}.$$

In addition, given $H_t$, $\bar{\Sigma}_{t,1}$, $\tilde{\Sigma}_{t,1}$ and $\bar{B}_{t,1}$ are constant. Thus

$$\begin{aligned}
\text{cov}\left[\mathbb{E}\left[\psi_{*,1} \,|\, H_t, \psi_{*,2}\right] | H_t\right] &= \text{cov}\left[\bar{\Sigma}_{t,1}\left(\Sigma_2^{-1}W_2\psi_{*,2} + \bar{B}_{t,1}\right) \Big| H_t\right], \\
&= \text{cov}\left[\bar{\Sigma}_{t,1}\Sigma_2^{-1}W_2\psi_{*,2} \,|\, H_t\right], \\
&= \bar{\Sigma}_{t,1}\Sigma_2^{-1}W_2\text{cov}\left[\psi_{*,2} \,|\, H_t\right]W_2^\top\Sigma_2^{-1}\bar{\Sigma}_{t,1}, \\
&= \bar{\Sigma}_{t,1}\Sigma_2^{-1}W_2\bar{\bar{\Sigma}}_{t,2}W_2^\top\Sigma_2^{-1}\bar{\Sigma}_{t,1}.
\end{aligned}$$

Finally, total covariance decomposition [Weiss, 2005, Hong et al., 2022b] leads to

$$\begin{aligned}
\bar{\bar{\Sigma}}_{t,1} = \text{cov}\left[\psi_{*,1} \,|\, H_t\right] &= \mathbb{E}\left[\text{cov}\left[\psi_{*,1} \,|\, H_t, \psi_{*,2}\right] | H_t\right] + \text{cov}\left[\mathbb{E}\left[\psi_{*,1} \,|\, H_t, \psi_{*,2}\right] | H_t\right], \\
&= \bar{\Sigma}_{t,1} + \bar{\Sigma}_{t,1}\Sigma_2^{-1}W_2\bar{\bar{\Sigma}}_{t,2}W_2^\top\Sigma_2^{-1}\bar{\Sigma}_{t,1}.
\end{aligned}$$

Now using the techniques, this can be generalized using the same technique as above to

$$\bar{\bar{\Sigma}}_{t,\ell} = \bar{\Sigma}_{t,\ell} + \bar{\Sigma}_{t,\ell}\Sigma_{\ell+1}^{-1}W_{\ell+1}\bar{\bar{\Sigma}}_{t,\ell+1}W_{\ell+1}^\top\Sigma_{\ell+1}^{-1}\bar{\Sigma}_{t,\ell}, \qquad \forall \ell \in [L-1].$$

Then, by induction, we get that

$$\bar{\bar{\Sigma}}_{t,1} = \sum_{\ell \in [L]} \bar{P}_\ell \bar{\Sigma}_{t,\ell} \bar{P}_\ell^\top \,, \qquad\qquad \forall \ell \in [L-1]\,,$$

where we use that by definition $\bar{\bar{\Sigma}}_{t,L} = \mathrm{cov}\left[\psi_{*,L} \,|\, H_t\right] = \bar{\Sigma}_{t,L}$ and set $\bar{P}_1 = I_d$ and $\bar{P}_\ell = \prod_{k=1}^{\ell-1} \bar{\Sigma}_{t,k} \Sigma_{k+1}^{-1} W_{k+1}$ for any $\ell \in [L]/\{1\}$. Plugging this in Eq. (31) leads to

$$\begin{aligned}
\check{\Sigma}_{t,i} &= \hat{\Sigma}_{t,i} + \sum_{\ell \in [L]} \hat{\Sigma}_{t,i} \Sigma_1^{-1} W_1 \bar{P}_\ell \bar{\Sigma}_{t,\ell} \bar{P}_\ell^\top W_1^\top \Sigma_1^{-1} \hat{\Sigma}_{t,i}\,, \\
&= \hat{\Sigma}_{t,i} + \sum_{\ell \in [L]} \hat{\Sigma}_{t,i} \Sigma_1^{-1} W_1 \bar{P}_\ell \bar{\Sigma}_{t,\ell} (\hat{\Sigma}_{t,i} \Sigma_1^{-1} W_1)^\top\,, \\
&= \hat{\Sigma}_{t,i} + \sum_{\ell \in [L]} P_{i,\ell} \bar{\Sigma}_{t,\ell} P_{i,\ell}^\top\,,
\end{aligned}$$

where $P_{i,\ell} = \hat{\Sigma}_{t,i} \Sigma_1^{-1} W_1 \bar{P}_\ell = \hat{\Sigma}_{t,i} \Sigma_1^{-1} W_1 \prod_{k=1}^{\ell-1} \bar{\Sigma}_{t,k} \Sigma_{k+1}^{-1} W_{k+1}$.

## D.4   Proof of lemma D.3

We prove this result by induction. We start with the base case when $\ell = 1$.

**(I) Base case.** Let $u = \sigma^{-1} \hat{\Sigma}_{t,A_t}^{\frac{1}{2}} X_t$ From the expression of $\bar{\Sigma}_{t,1}$ in Eq. (18), we have that

$$\begin{aligned}
\bar{\Sigma}_{t+1,1}^{-1} - \bar{\Sigma}_{t,1}^{-1} &= W_1^\top \left( \Sigma_1^{-1} - \Sigma_1^{-1}(\hat{\Sigma}_{t,A_t}^{-1} + \sigma^{-2} X_t X_t^\top)^{-1} \Sigma_1^{-1} - (\Sigma_1^{-1} - \Sigma_1^{-1} \hat{\Sigma}_{t,A_t} \Sigma_1^{-1}) \right) W_1\,, \\
&= W_1^\top \left( \Sigma_1^{-1}(\hat{\Sigma}_{t,A_t} - (\hat{\Sigma}_{t,A_t}^{-1} + \sigma^{-2} X_t X_t^\top)^{-1}) \Sigma_1^{-1} \right) W_1\,, \\
&= W_1^\top \left( \Sigma_1^{-1} \hat{\Sigma}_{t,A_t}^{\frac{1}{2}} (I_d - (I_d + \sigma^{-2} \hat{\Sigma}_{t,A_t}^{\frac{1}{2}} X_t X_t^\top \hat{\Sigma}_{t,A_t}^{\frac{1}{2}})^{-1}) \hat{\Sigma}_{t,A_t}^{\frac{1}{2}} \Sigma_1^{-1} \right) W_1\,, \\
&= W_1^\top \left( \Sigma_1^{-1} \hat{\Sigma}_{t,A_t}^{\frac{1}{2}} (I_d - (I_d + u u^\top)^{-1}) \hat{\Sigma}_{t,A_t}^{\frac{1}{2}} \Sigma_1^{-1} \right) W_1\,, \\
&\overset{(i)}{=} W_1^\top \left( \Sigma_1^{-1} \hat{\Sigma}_{t,A_t}^{\frac{1}{2}} \frac{u u^\top}{1 + u^\top u} \hat{\Sigma}_{t,A_t}^{\frac{1}{2}} \Sigma_1^{-1} \right) W_1\,, \\
&\overset{(ii)}{=} \sigma^{-2} W_1^\top \Sigma_1^{-1} \hat{\Sigma}_{t,A_t} \frac{X_t X_t^\top}{1 + u^\top u} \hat{\Sigma}_{t,A_t} \Sigma_1^{-1} W_1\,.
\end{aligned} \tag{32}$$

In $(i)$ we use the Sherman-Morrison formula. Note that $(ii)$ says that $\bar{\Sigma}_{t+1,1}^{-1} - \bar{\Sigma}_{t,1}^{-1}$ is one-rank which we will also need in induction step. Now, we have that $\|X_t\|^2 = 1$. Therefore,

$$1 + u^\top u = 1 + \sigma^{-2} X_t^\top \hat{\Sigma}_{t,A_t} X_t \leq 1 + \sigma^{-2} \lambda_1(\Sigma_1) \|X_t\|^2 = 1 + \sigma^{-2} \sigma_1^2 \leq \sigma_{\mathrm{MAX}}^2\,,$$

where we use that by definition of $\sigma_{\mathrm{MAX}}^2$ in Lemma D.3, we have that $\sigma_{\mathrm{MAX}}^2 \geq 1 + \sigma^{-2} \sigma_1^2$. Therefore, by taking the inverse, we get that $\frac{1}{1+u^\top u} \geq \sigma_{\mathrm{MAX}}^{-2}$. Combining this with Eq. (32) leads to

$$\bar{\Sigma}_{t+1,1}^{-1} - \bar{\Sigma}_{t,1}^{-1} \succeq \sigma^{-2} \sigma_{\mathrm{MAX}}^{-2} W_1^\top \Sigma_1^{-1} \hat{\Sigma}_{t,A_t} X_t X_t^\top \hat{\Sigma}_{t,A_t} \Sigma_1^{-1} W_1$$

Noticing that $P_{A_t,1} = \hat{\Sigma}_{t,A_t} \Sigma_1^{-1} W_1$ concludes the proof of the base case when $\ell = 1$.

**(II) Induction step.** Let $\ell \in [L]/\{1\}$ and suppose that $\bar{\Sigma}_{t+1,\ell-1}^{-1} - \bar{\Sigma}_{t,\ell-1}^{-1}$ is one-rank and that it holds for $\ell - 1$ that

$$\bar{\Sigma}_{t+1,\ell-1}^{-1} - \bar{\Sigma}_{t,\ell-1}^{-1} \succeq \sigma^{-2} \sigma_{\mathrm{MAX}}^{-2(\ell-1)} P_{A_t,\ell-1}^\top X_t X_t^\top P_{A_t,\ell-1}\,, \quad \text{where } \sigma_{\mathrm{MAX}}^{-2} = \max_{\ell \in [L]} 1 + \sigma^{-2} \sigma_\ell^2.$$

Then, we want to show that $\bar{\Sigma}_{t+1,\ell}^{-1} - \bar{\Sigma}_{t,\ell}^{-1}$ is also one-rank and that it holds that

$$\bar{\Sigma}_{t+1,\ell}^{-1} - \bar{\Sigma}_{t,\ell}^{-1} \succeq \sigma^{-2} \sigma_{\mathrm{MAX}}^{-2\ell} P_{A_t,\ell}^\top X_t X_t^\top P_{A_t,\ell}\,, \qquad \text{where } \sigma_{\mathrm{MAX}}^{-2} = \max_{\ell \in [L]} 1 + \sigma^{-2} \sigma_\ell^2.$$

This is achieved as follows. First, we notice that by the induction hypothesis, we have that $\tilde{\Sigma}_{t+1,\ell-1}^{-1} - \bar{G}_{t,\ell-1} = \bar{\Sigma}_{t+1,\ell-1}^{-1} - \bar{\Sigma}_{t,\ell-1}^{-1}$ is one-rank. In addition, the matrix is positive semi-definite. Thus we can write it as $\tilde{\Sigma}_{t+1,\ell-1}^{-1} - \bar{G}_{t,\ell-1} = uu^\top$ where $u \in \mathbb{R}^d$. Then, similarly to the base case, we have

$$
\begin{aligned}
\bar{\Sigma}_{t+1,\ell}^{-1} - \bar{\Sigma}_{t,\ell}^{-1} &= \tilde{\Sigma}_{t+1,\ell}^{-1} - \tilde{\Sigma}_{t,\ell}^{-1}\,, \\
&= W_\ell^\top \left(\Sigma_\ell + \tilde{\Sigma}_{t+1,\ell-1}\right)^{-1} W_\ell - W_\ell^\top \left(\Sigma_\ell + \tilde{\Sigma}_{t,\ell-1}\right)^{-1} W_\ell\,, \\
&= W_\ell^\top \left[\left(\Sigma_\ell + \tilde{\Sigma}_{t+1,\ell-1}\right)^{-1} - \left(\Sigma_\ell + \tilde{\Sigma}_{t,\ell-1}\right)^{-1}\right] W_\ell\,, \\
&= W_\ell^\top \Sigma_\ell^{-1} \left[\left(\Sigma_\ell^{-1} + \bar{G}_{t,\ell-1}\right)^{-1} - \left(\Sigma_\ell^{-1} + \tilde{\Sigma}_{t+1,\ell-1}^{-1}\right)^{-1}\right] \Sigma_\ell^{-1} W_\ell\,, \\
&= W_\ell^\top \Sigma_\ell^{-1} \left[\left(\Sigma_\ell^{-1} + \bar{G}_{t,\ell-1}\right)^{-1} - \left(\Sigma_\ell^{-1} + \bar{G}_{t,\ell-1} + \tilde{\Sigma}_{t+1,\ell-1}^{-1} - \bar{G}_{t,\ell-1}\right)^{-1}\right] \Sigma_\ell^{-1} W_\ell\,, \\
&= W_\ell^\top \Sigma_\ell^{-1} \left[\left(\Sigma_\ell^{-1} + \bar{G}_{t,\ell-1}\right)^{-1} - \left(\Sigma_\ell^{-1} + \bar{G}_{t,\ell-1} + uu^\top\right)^{-1}\right] \Sigma_\ell^{-1} W_\ell\,, \\
&= W_\ell^\top \Sigma_\ell^{-1} \left[\bar{\Sigma}_{t,\ell-1} - \left(\bar{\Sigma}_{t,\ell-1}^{-1} + uu^\top\right)^{-1}\right] \Sigma_\ell^{-1} W_\ell\,, \\
&= W_\ell^\top \Sigma_\ell^{-1} \left[\bar{\Sigma}_{t,\ell-1} \frac{uu^\top}{1 + u^\top \bar{\Sigma}_{t,\ell-1} u} \bar{\Sigma}_{t,\ell-1}\right] \Sigma_\ell^{-1} W_\ell\,, \\
&= W_\ell^\top \Sigma_\ell^{-1} \bar{\Sigma}_{t,\ell-1} \frac{uu^\top}{1 + u^\top \bar{\Sigma}_{t,\ell-1} u} \bar{\Sigma}_{t,\ell-1} \Sigma_\ell^{-1} W_\ell
\end{aligned}
$$

However, we it follows from the induction hypothesis that $uu^\top = \tilde{\Sigma}_{t+1,\ell-1}^{-1} - \bar{G}_{t,\ell-1} = \bar{\Sigma}_{t+1,\ell-1}^{-1} - \bar{\Sigma}_{t,\ell-1}^{-1} \succeq \sigma^{-2} \sigma_{\text{MAX}}^{-2(\ell-1)} P_{A_t,\ell-1}^\top X_t X_t^\top P_{A_t,\ell-1}$. Therefore,

$$
\begin{aligned}
\bar{\Sigma}_{t+1,\ell}^{-1} - \bar{\Sigma}_{t,\ell}^{-1} &= W_\ell^\top \Sigma_\ell^{-1} \bar{\Sigma}_{t,\ell-1} \frac{uu^\top}{1 + u^\top \bar{\Sigma}_{t,\ell-1} u} \bar{\Sigma}_{t,\ell-1} \Sigma_\ell^{-1} W_\ell\,, \\
&\succeq W_\ell^\top \Sigma_\ell^{-1} \bar{\Sigma}_{t,\ell-1} \frac{\sigma^{-2} \sigma_{\text{MAX}}^{-2(\ell-1)} P_{A_t,\ell-1}^\top X_t X_t^\top P_{A_t,\ell-1}}{1 + u^\top \bar{\Sigma}_{t,\ell-1} u} \bar{\Sigma}_{t,\ell-1} \Sigma_\ell^{-1} W_\ell\,, \\
&= \frac{\sigma^{-2} \sigma_{\text{MAX}}^{-2(\ell-1)}}{1 + u^\top \bar{\Sigma}_{t,\ell-1} u} W_\ell^\top \Sigma_\ell^{-1} \bar{\Sigma}_{t,\ell-1} P_{A_t,\ell-1}^\top X_t X_t^\top P_{A_t,\ell-1} \bar{\Sigma}_{t,\ell-1} \Sigma_\ell^{-1} W_\ell\,, \\
&= \frac{\sigma^{-2} \sigma_{\text{MAX}}^{-2(\ell-1)}}{1 + u^\top \bar{\Sigma}_{t,\ell-1} u} P_{A_t,\ell}^\top X_t X_t^\top P_{A_t,\ell}\,.
\end{aligned}
$$

Finally, we use that $1 + u^\top \bar{\Sigma}_{t,\ell-1} u \leq 1 + \|u\|_2 \lambda_1(\bar{\Sigma}_{t,\ell-1}) \leq 1 + \sigma^{-2} \sigma_\ell^2$. Here we use that $\|u\|_2 \leq \sigma^{-2}$, which can also be proven by induction, and that $\lambda_1(\bar{\Sigma}_{t,\ell-1}) \leq \sigma_\ell^2$, which follows from the expression of $\bar{\Sigma}_{t,\ell-1}$ in Section 3.1. Therefore, we have that

$$
\begin{aligned}
\bar{\Sigma}_{t+1,\ell}^{-1} - \bar{\Sigma}_{t,\ell}^{-1} &\succeq \frac{\sigma^{-2} \sigma_{\text{MAX}}^{-2(\ell-1)}}{1 + u^\top \bar{\Sigma}_{t,\ell-1} u} P_{A_t,\ell}^\top X_t X_t^\top P_{A_t,\ell}\,, \\
&\succeq \frac{\sigma^{-2} \sigma_{\text{MAX}}^{-2(\ell-1)}}{1 + \sigma^{-2} \sigma_\ell^2} P_{A_t,\ell}^\top X_t X_t^\top P_{A_t,\ell}\,, \\
&\succeq \sigma^{-2} \sigma_{\text{MAX}}^{-2\ell} P_{A_t,\ell}^\top X_t X_t^\top P_{A_t,\ell}\,,
\end{aligned}
$$

where the last inequality follows from the definition of $\sigma_{\text{MAX}}^2 = \max_{\ell \in [L]} 1 + \sigma^{-2} \sigma_\ell^2$. This concludes the proof.

### D.5 Proof of theorem 4.1

We start with the following standard result which we borrow from [Hong et al., 2022a, Aouali et al., 2023b],

$$
\mathcal{BR}(n) \leq \sqrt{2n \log(1/\delta)} \sqrt{\mathbb{E}\left[\sum_{t=1}^n \|X_t\|_{\tilde{\Sigma}_{t,A_t}}^2\right]} + cn\delta\,, \qquad \text{where } c > 0 \text{ is a constant}\,. \tag{33}
$$

Then we use Lemma D.2 and express the marginal covariance $\check{\Sigma}_{t,A_t}$ as

$$\check{\Sigma}_{t,i} = \hat{\Sigma}_{t,i} + \sum_{\ell \in [L]} \mathrm{P}_{i,\ell} \bar{\Sigma}_{t,\ell} \mathrm{P}_{i,\ell}^\top, \qquad \text{where } \mathrm{P}_{i,\ell} = \hat{\Sigma}_{t,i} \Sigma_1^{-1} \mathrm{W}_1 \prod_{k=1}^{\ell-1} \bar{\Sigma}_{t,k} \Sigma_{k+1}^{-1} \mathrm{W}_{k+1}. \tag{34}$$

Therefore, we can decompose $\|X_t\|_{\check{\Sigma}_{t,A_t}}^2$ as

$$\|X_t\|_{\check{\Sigma}_{t,A_t}}^2 = \sigma^2 \frac{X_t^\top \check{\Sigma}_{t,A_t} X_t}{\sigma^2} \overset{(i)}{=} \sigma^2 \big( \sigma^{-2} X_t^\top \hat{\Sigma}_{t,A_t} X_t + \sigma^{-2} \sum_{\ell \in [L]} X_t^\top \mathrm{P}_{A_t,\ell} \bar{\Sigma}_{t,\ell} \mathrm{P}_{A_t,\ell}^\top X_t \big),$$

$$\overset{(ii)}{\le} c_0 \log(1 + \sigma^{-2} X_t^\top \hat{\Sigma}_{t,A_t} X_t) + \sum_{\ell \in [L]} c_\ell \log(1 + \sigma^{-2} X_t^\top \mathrm{P}_{A_t,\ell} \bar{\Sigma}_{t,\ell} \mathrm{P}_{A_t,\ell}^\top X_t), \tag{35}$$

where $(i)$ follows from Eq. (34), and we use the following inequality in $(ii)$

$$x = \frac{x}{\log(1+x)} \log(1+x) \le \left( \max_{x \in [0,u]} \frac{x}{\log(1+x)} \right) \log(1+x) = \frac{u}{\log(1+u)} \log(1+x),$$

which holds for any $x \in [0, u]$, where constants $c_0$ and $c_\ell$ are derived as

$$c_0 = \frac{\sigma_1^2}{\log(1 + \frac{\sigma_1^2}{\sigma^2})}, \quad c_\ell = \frac{\sigma_{\ell+1}^2}{\log(1 + \frac{\sigma_{\ell+1}^2}{\sigma^2})}, \text{ with the convention that } \sigma_{L+1} = 1.$$

The derivation of $c_0$ uses that

$$X_t^\top \hat{\Sigma}_{t,A_t} X_t \le \lambda_1(\hat{\Sigma}_{t,A_t}) \|X_t\|^2 \le \lambda_d^{-1}(\Sigma_1^{-1} + G_{t,A_t}) \le \lambda_d^{-1}(\Sigma_1^{-1}) = \lambda_1(\Sigma_1) = \sigma_1^2.$$

The derivation of $c_\ell$ follows from

$$X_t^\top \mathrm{P}_{A_t,\ell} \bar{\Sigma}_{t,\ell} \mathrm{P}_{A_t,\ell}^\top X_t \le \lambda_1(\mathrm{P}_{A_t,\ell} \mathrm{P}_{A_t,\ell}^\top) \lambda_1(\bar{\Sigma}_{t,\ell}) \|X_t\|^2 \le \sigma_{\ell+1}^2.$$

Therefore, from Eq. (35) and Eq. (33), we get that

$$\mathcal{BR}(n) \le \sqrt{2n\log(1/\delta)} \Big( \mathbb{E}\Big[ c_0 \sum_{t=1}^n \log(1 + \sigma^{-2} X_t^\top \hat{\Sigma}_{t,A_t} X_t)$$

$$+ \sum_{\ell \in [L]} c_\ell \sum_{t=1}^n \log(1 + \sigma^{-2} X_t^\top \mathrm{P}_{A_t,\ell} \bar{\Sigma}_{t,\ell} \mathrm{P}_{A_t,\ell}^\top X_t) \Big] \Big)^{\frac{1}{2}} + cn\delta \tag{36}$$

Now we focus on bounding the logarithmic terms in Eq. (36).

**(I) First term in Eq. (36)** We first rewrite this term as

$$\log(1 + \sigma^{-2} X_t^\top \hat{\Sigma}_{t,A_t} X_t) \overset{(i)}{=} \log\det(I_d + \sigma^{-2} \hat{\Sigma}_{t,A_t}^{\frac{1}{2}} X_t X_t^\top \hat{\Sigma}_{t,A_t}^{\frac{1}{2}}),$$

$$= \log\det(\hat{\Sigma}_{t,A_t}^{-1} + \sigma^{-2} X_t X_t^\top) - \log\det(\hat{\Sigma}_{t,A_t}^{-1}) = \log\det(\hat{\Sigma}_{t+1,A_t}^{-1}) - \log\det(\hat{\Sigma}_{t,A_t}^{-1}),$$

where $(i)$ follows from the Weinstein–Aronszajn identity. Then we sum over all rounds $t \in [n]$, and get a telescoping

$$\sum_{t=1}^n \log\det(I_d + \sigma^{-2} \hat{\Sigma}_{t,A_t}^{\frac{1}{2}} X_t X_t^\top \hat{\Sigma}_{t,A_t}^{\frac{1}{2}}) = \sum_{t=1}^n \log\det(\hat{\Sigma}_{t+1,A_t}^{-1}) - \log\det(\hat{\Sigma}_{t,A_t}^{-1}),$$

$$= \sum_{t=1}^n \sum_{i=1}^K \log\det(\hat{\Sigma}_{t+1,i}^{-1}) - \log\det(\hat{\Sigma}_{t,i}^{-1}) = \sum_{i=1}^K \sum_{t=1}^n \log\det(\hat{\Sigma}_{t+1,i}^{-1}) - \log\det(\hat{\Sigma}_{t,i}^{-1}),$$

$$= \sum_{i=1}^K \log\det(\hat{\Sigma}_{n+1,i}^{-1}) - \log\det(\hat{\Sigma}_{1,i}^{-1}) \overset{(i)}{=} \sum_{i=1}^K \log\det(\Sigma_1^{\frac{1}{2}} \hat{\Sigma}_{n+1,i}^{-1} \Sigma_1^{\frac{1}{2}}),$$

where $(i)$ follows from the fact that $\hat{\Sigma}_{1,i} = \Sigma_1$. Now we use the inequality of arithmetic and geometric means and get

$$
\sum_{t=1}^{n} \log \det(I_d + \sigma^{-2}\hat{\Sigma}_{t,A_t}^{\frac{1}{2}} X_t X_t^{\top} \hat{\Sigma}_{t,A_t}^{\frac{1}{2}}) = \sum_{i=1}^{K} \log \det(\Sigma_1^{\frac{1}{2}} \hat{\Sigma}_{n+1,i}^{-1} \Sigma_1^{\frac{1}{2}}) ,
$$

$$
\leq \sum_{i=1}^{K} d \log\left( \frac{1}{d} \operatorname{Tr}(\Sigma_1^{\frac{1}{2}} \hat{\Sigma}_{n+1,i}^{-1} \Sigma_1^{\frac{1}{2}}) \right) , \qquad (37)
$$

$$
\leq \sum_{i=1}^{K} d \log\left( 1 + \frac{n}{d}\frac{\sigma_1^2}{\sigma^2} \right) = Kd \log\left( 1 + \frac{n}{d}\frac{\sigma_1^2}{\sigma^2} \right) .
$$

**(II) Remaining terms in Eq. (36)** Let $\ell \in [L]$. Then we have that

$$
\log(1 + \sigma^{-2} X_t^{\top} P_{A_t,\ell} \bar{\Sigma}_{t,\ell} P_{A_t,\ell}^{\top} X_t) = \sigma_{\mathrm{MAX}}^{2\ell} \sigma_{\mathrm{MAX}}^{-2\ell} \log(1 + \sigma^{-2} X_t^{\top} P_{A_t,\ell} \bar{\Sigma}_{t,\ell} P_{A_t,\ell}^{\top} X_t) ,
$$

$$
\leq \sigma_{\mathrm{MAX}}^{2\ell} \log(1 + \sigma^{-2} \sigma_{\mathrm{MAX}}^{-2\ell} X_t^{\top} P_{A_t,\ell} \bar{\Sigma}_{t,\ell} P_{A_t,\ell}^{\top} X_t) ,
$$

$$
\overset{(i)}{=} \sigma_{\mathrm{MAX}}^{2\ell} \log \det(I_d + \sigma^{-2} \sigma_{\mathrm{MAX}}^{-2\ell} \bar{\Sigma}_{t,\ell}^{\frac{1}{2}} P_{A_t,\ell}^{\top} X_t X_t^{\top} P_{A_t,\ell} \bar{\Sigma}_{t,\ell}^{\frac{1}{2}}) ,
$$

$$
= \sigma_{\mathrm{MAX}}^{2\ell} \Big( \log \det(\bar{\Sigma}_{t,\ell}^{-1} + \sigma^{-2} \sigma_{\mathrm{MAX}}^{-2\ell} P_{A_t,\ell}^{\top} X_t X_t^{\top} P_{A_t,\ell}) - \log \det(\bar{\Sigma}_{t,\ell}^{-1}) \Big) ,
$$

where we use the Weinstein–Aronszajn identity in $(i)$. Now we know from Lemma D.3 that the following inequality holds $\sigma^{-2} \sigma_{\mathrm{MAX}}^{-2\ell} P_{A_t,\ell}^{\top} X_t X_t^{\top} P_{A_t,\ell} \preceq \bar{\Sigma}_{t+1,\ell}^{-1} - \bar{\Sigma}_{t,\ell}^{-1}$. As a result, we get that $\bar{\Sigma}_{t,\ell}^{-1} + \sigma^{-2} \sigma_{\mathrm{MAX}}^{-2\ell} P_{A_t,\ell}^{\top} X_t X_t^{\top} P_{A_t,\ell} \preceq \bar{\Sigma}_{t+1,\ell}^{-1}$. Thus,

$$
\log(1 + \sigma^{-2} X_t^{\top} P_{A_t,\ell} \bar{\Sigma}_{t,\ell} P_{A_t,\ell}^{\top} X_t) \leq \sigma_{\mathrm{MAX}}^{2\ell} \Big( \log \det(\bar{\Sigma}_{t+1,\ell}^{-1}) - \log \det(\bar{\Sigma}_{t,\ell}^{-1}) \Big) ,
$$

Then we sum over all rounds $t \in [n]$, and get a telescoping

$$
\sum_{t=1}^{n} \log(1 + \sigma^{-2} X_t^{\top} P_{A_t,\ell} \bar{\Sigma}_{t,\ell} P_{A_t,\ell}^{\top} X_t) \leq \sigma_{\mathrm{MAX}}^{2\ell} \sum_{t=1}^{n} \log \det(\bar{\Sigma}_{t+1,\ell}^{-1}) - \log \det(\bar{\Sigma}_{t,\ell}^{-1}) ,
$$

$$
= \sigma_{\mathrm{MAX}}^{2\ell} \Big( \log \det(\bar{\Sigma}_{n+1,\ell}^{-1}) - \log \det(\bar{\Sigma}_{1,\ell}^{-1}) \Big) ,
$$

$$
\overset{(i)}{=} \sigma_{\mathrm{MAX}}^{2\ell} \Big( \log \det(\bar{\Sigma}_{n+1,\ell}^{-1}) - \log \det(\Sigma_{\ell+1}^{-1}) \Big) ,
$$

$$
= \sigma_{\mathrm{MAX}}^{2\ell} \Big( \log \det(\Sigma_{\ell+1}^{\frac{1}{2}} \bar{\Sigma}_{n+1,\ell}^{-1} \Sigma_{\ell+1}^{\frac{1}{2}}) \Big) ,
$$

where we use that $\bar{\Sigma}_{1,\ell} = \Sigma_{\ell+1}$ in $(i)$. Finally, we use the inequality of arithmetic and geometric means and get that

$$
\sum_{t=1}^{n} \log(1 + \sigma^{-2} X_t^{\top} P_{A_t,\ell} \bar{\Sigma}_{t,\ell} P_{A_t,\ell}^{\top} X_t) \leq \sigma_{\mathrm{MAX}}^{2\ell} \Big( \log \det(\Sigma_{\ell+1}^{\frac{1}{2}} \bar{\Sigma}_{n+1,\ell}^{-1} \Sigma_{\ell+1}^{\frac{1}{2}}) \Big) ,
$$

$$
\leq d\sigma_{\mathrm{MAX}}^{2\ell} \log\left( \frac{1}{d} \operatorname{Tr}(\Sigma_{\ell+1}^{\frac{1}{2}} \bar{\Sigma}_{n+1,\ell}^{-1} \Sigma_{\ell+1}^{\frac{1}{2}}) \right) , \qquad (38)
$$

$$
\leq d\sigma_{\mathrm{MAX}}^{2\ell} \log\left( 1 + \frac{\sigma_{\ell+1}^2}{\sigma_\ell^2} \right) ,
$$

The last inequality follows from the expression of $\bar{\Sigma}_{n+1,\ell}^{-1}$ in Eq. (18) that leads to

$$
\Sigma_{\ell+1}^{\frac{1}{2}} \bar{\Sigma}_{n+1,\ell}^{-1} \Sigma_{\ell+1}^{\frac{1}{2}} = I_d + \Sigma_{\ell+1}^{\frac{1}{2}} \bar{G}_{t,\ell} \Sigma_{\ell+1}^{\frac{1}{2}} ,
$$

$$
= I_d + \Sigma_{\ell+1}^{\frac{1}{2}} W_\ell^{\top} \big( \Sigma_\ell^{-1} - \Sigma_\ell^{-1} \bar{\Sigma}_{t,\ell-1} \Sigma_\ell^{-1} \big) W_\ell \Sigma_{\ell+1}^{\frac{1}{2}} , \qquad (39)
$$

since $\bar{G}_{t,\ell} = W_\ell^\top \left(\Sigma_\ell^{-1} - \Sigma_\ell^{-1}\bar{\Sigma}_{t,\ell-1}\Sigma_\ell^{-1}\right)W_\ell$. This allows us to bound $\frac{1}{d}\operatorname{Tr}(\Sigma_{\ell+1}^{\frac{1}{2}}\bar{\Sigma}_{n+1,\ell}^{-1}\Sigma_{\ell+1}^{\frac{1}{2}})$ as

$$
\begin{aligned}
\frac{1}{d}\operatorname{Tr}(\Sigma_{\ell+1}^{\frac{1}{2}}\bar{\Sigma}_{n+1,\ell}^{-1}\Sigma_{\ell+1}^{\frac{1}{2}}) &= \frac{1}{d}\operatorname{Tr}(I_d + \Sigma_{\ell+1}^{\frac{1}{2}}W_\ell^\top\left(\Sigma_\ell^{-1} - \Sigma_\ell^{-1}\bar{\Sigma}_{t,\ell-1}\Sigma_\ell^{-1}\right)W_\ell\Sigma_{\ell+1}^{\frac{1}{2}})\,, \\
&= \frac{1}{d}(d + \operatorname{Tr}(\Sigma_{\ell+1}^{\frac{1}{2}}W_\ell^\top\left(\Sigma_\ell^{-1} - \Sigma_\ell^{-1}\bar{\Sigma}_{t,\ell-1}\Sigma_\ell^{-1}\right)W_\ell\Sigma_{\ell+1}^{\frac{1}{2}})\,, \\
&\leq 1 + \frac{1}{d}\sum_{k=1}^{d}\lambda_1(\Sigma_{\ell+1}^{\frac{1}{2}}W_\ell^\top\left(\Sigma_\ell^{-1} - \Sigma_\ell^{-1}\bar{\Sigma}_{t,\ell-1}\Sigma_\ell^{-1}\right)W_\ell\Sigma_{\ell+1}^{\frac{1}{2}}\,, \\
&\leq 1 + \frac{1}{d}\sum_{k=1}^{d}\lambda_1(\Sigma_{\ell+1})\lambda_1(W_\ell^\top W_\ell)\lambda_1\left(\Sigma_\ell^{-1} - \Sigma_\ell^{-1}\bar{\Sigma}_{t,\ell-1}\Sigma_\ell^{-1}\right)\,, \\
&\leq 1 + \frac{1}{d}\sum_{k=1}^{d}\lambda_1(\Sigma_{\ell+1})\lambda_1(W_\ell^\top W_\ell)\lambda_1\left(\Sigma_\ell^{-1}\right)\,, \\
&\leq 1 + \frac{1}{d}\sum_{k=1}^{d}\frac{\sigma_{\ell+1}^2}{\sigma_\ell^2} = 1 + \frac{\sigma_{\ell+1}^2}{\sigma_\ell^2}\,, \quad (40)
\end{aligned}
$$

where we use the assumption that $\lambda_1(W_\ell^\top W_\ell) = 1$ **(A2)** and that $\lambda_1(\Sigma_{\ell+1}) = \sigma_{\ell+1}^2$ and $\lambda_1(\Sigma_\ell^{-1}) = 1/\sigma_\ell^2$. This is because $\Sigma_\ell = \sigma_\ell^2 I_d$ for any $\ell \in [L+1]$. Finally, plugging Eqs. (37) and (38) in Eq. (36) concludes the proof.

### D.6   Proof of proposition 4.2

We use exactly the same proof in Appendix D.5, with one change to account for the sparsity assumption **(A3)**. The change corresponds to Eq. (38). First, recall that Eq. (38) writes

$$
\sum_{t=1}^{n}\log(1 + \sigma^{-2}X_t^\top P_{A_t,\ell}\bar{\Sigma}_{t,\ell}P_{A_t,\ell}^\top X_t) \leq \sigma_{\text{MAX}}^{2\ell}\left(\log\det(\Sigma_{\ell+1}^{\frac{1}{2}}\bar{\Sigma}_{n+1,\ell}^{-1}\Sigma_{\ell+1}^{\frac{1}{2}})\right)\,,
$$

where

$$
\begin{aligned}
\Sigma_{\ell+1}^{\frac{1}{2}}\bar{\Sigma}_{n+1,\ell}^{-1}\Sigma_{\ell+1}^{\frac{1}{2}} &= I_d + \Sigma_{\ell+1}^{\frac{1}{2}}W_\ell^\top\left(\Sigma_\ell^{-1} - \Sigma_\ell^{-1}\bar{\Sigma}_{t,\ell-1}\Sigma_\ell^{-1}\right)W_\ell\Sigma_{\ell+1}^{\frac{1}{2}}\,, \\
&= I_d + \sigma_{\ell+1}^2 W_\ell^\top\left(\Sigma_\ell^{-1} - \Sigma_\ell^{-1}\bar{\Sigma}_{t,\ell-1}\Sigma_\ell^{-1}\right)W_\ell\,, \quad (41)
\end{aligned}
$$

where the second equality follows from the assumption that $\Sigma_{\ell+1} = \sigma_{\ell+1}^2 I_d$. But notice that in our assumption, **(A3)**, we assume that $W_\ell = (\bar{W}_\ell, 0_{d,d-d_\ell})$, where $\bar{W}_\ell \in \mathbb{R}^{d\times d_\ell}$ for any $\ell \in [L]$. Therefore, we have that for any $d \times d$ matrix $B \in \mathbb{R}^{dd\times d}$, the following holds, $W_\ell^\top B W_\ell = \begin{pmatrix} \bar{W}_\ell^\top B \bar{W}_\ell & 0_{d_\ell,d-d_\ell} \\ 0_{d-d_\ell,d_\ell} & 0_{d-d_\ell,d-d_\ell} \end{pmatrix}$. In particular, we have that

$$
W_\ell^\top\left(\Sigma_\ell^{-1} - \Sigma_\ell^{-1}\bar{\Sigma}_{t,\ell-1}\Sigma_\ell^{-1}\right)W_\ell = \begin{pmatrix} \bar{W}_\ell^\top\left(\Sigma_\ell^{-1} - \Sigma_\ell^{-1}\bar{\Sigma}_{t,\ell-1}\Sigma_\ell^{-1}\right)\bar{W}_\ell & 0_{d_\ell,d-d_\ell} \\ 0_{d-d_\ell,d_\ell} & 0_{d-d_\ell,d-d_\ell} \end{pmatrix}\,. \quad (42)
$$

Therefore, plugging this in Eq. (41) yields that

$$
\Sigma_{\ell+1}^{\frac{1}{2}}\bar{\Sigma}_{n+1,\ell}^{-1}\Sigma_{\ell+1}^{\frac{1}{2}} = \begin{pmatrix} I_{d_\ell} + \sigma_{\ell+1}^2\bar{W}_\ell^\top\left(\Sigma_\ell^{-1} - \Sigma_\ell^{-1}\bar{\Sigma}_{t,\ell-1}\Sigma_\ell^{-1}\right)\bar{W}_\ell & 0_{d_\ell,d-d_\ell} \\ 0_{d-d_\ell,d_\ell} & I_{d-d_\ell} \end{pmatrix}\,. \quad (43)
$$

As a result, $\det(\Sigma_{\ell+1}^{\frac{1}{2}}\bar{\Sigma}_{n+1,\ell}^{-1}\Sigma_{\ell+1}^{\frac{1}{2}}) = \det(I_{d_\ell} + \sigma_{\ell+1}^2\bar{W}_\ell^\top\left(\Sigma_\ell^{-1} - \Sigma_\ell^{-1}\bar{\Sigma}_{t,\ell-1}\Sigma_\ell^{-1}\right)\bar{W}_\ell)$. This allows us to move the problem from a $d$-dimensional one to a $d_\ell$-dimensional one. Then we use the inequality

of arithmetic and geometric means and get that

$$
\begin{aligned}
\sum_{t=1}^{n} \log(1 + \sigma^{-2} X_t^\top \mathrm{P}_{A_t,\ell} \bar{\Sigma}_{t,\ell} \mathrm{P}_{A_t,\ell}^\top X_t) &\leq \sigma_{\text{MAX}}^{2\ell}\Big( \log\det(\Sigma_{\ell+1}^{\frac{1}{2}} \bar{\Sigma}_{n+1,\ell}^{-1} \Sigma_{\ell+1}^{\frac{1}{2}}) \Big), \\
&= \sigma_{\text{MAX}}^{2\ell} \log\det(I_{d_\ell} + \sigma_{\ell+1}^2 \bar{\mathrm{W}}_\ell^\top \big(\Sigma_\ell^{-1} - \Sigma_\ell^{-1}\bar{\Sigma}_{t,\ell-1}\Sigma_\ell^{-1}\big)\bar{\mathrm{W}}_\ell), \\
&\leq d_\ell \sigma_{\text{MAX}}^{2\ell} \log\left( \frac{1}{d_\ell} \operatorname{Tr}(I_{d_\ell} + \sigma_{\ell+1}^2 \bar{\mathrm{W}}_\ell^\top \big(\Sigma_\ell^{-1} - \Sigma_\ell^{-1}\bar{\Sigma}_{t,\ell-1}\Sigma_\ell^{-1}\big)\bar{\mathrm{W}}_\ell) \right), \\
&\leq d_\ell \sigma_{\text{MAX}}^{2\ell} \log\left( 1 + \frac{\sigma_{\ell+1}^2}{\sigma_\ell^2} \right).
\end{aligned}
\tag{44}
$$

To get the last inequality, we use derivations similar to the ones we used in Eq. (40). Finally, the desired result in obtained by replacing Eq. (38) by Eq. (44) in the previous proof in Appendix D.5.

### D.7 Additional discussion: link to two-level hierarchies

The linear diffusion (15) can be marginalized into a 2-level hierarchy using two different strategies. The first one yields,

$$
\begin{aligned}
\psi_{*,L} &\sim \mathcal{N}(0, \sigma_{L+1}^2 \mathrm{B}_L \mathrm{B}_L^\top), \\
\theta_{*,i} \mid \psi_{*,L} &\sim \mathcal{N}(\psi_{*,L}, \Omega_1), \qquad\qquad \forall i \in [K],
\end{aligned}
\tag{45}
$$

with $\Omega_1 = \sigma_1^2 I_d + \sum_{\ell=1}^{L-1} \sigma_{\ell+1}^2 \mathrm{B}_\ell \mathrm{B}_\ell^\top$ and $\mathrm{B}_\ell = \prod_{k=1}^{\ell} \mathrm{W}_k$. The second strategy yields,

$$
\begin{aligned}
\psi_{*,1} &\sim \mathcal{N}(0, \Omega_2), \\
\theta_{*,i} \mid \psi_{*,1} &\sim \mathcal{N}(\psi_{*,1}, \sigma_1^2 I_d), \qquad\qquad \forall i \in [K],
\end{aligned}
\tag{46}
$$

where $\Omega_2 = \sum_{\ell=1}^{L} \sigma_{\ell+1}^2 \mathrm{B}_\ell \mathrm{B}_\ell^\top$. Recently, HierTS [Hong et al., 2022b] was developed for such two-level graphical models, and we call HierTS under (45) by HierTS-1 and HierTS under (46) by HierTS-2. Then, we start by highlighting the differences between these two variants of HierTS. First, their regret bounds scale as

$$
\texttt{HierTS-1}: \tilde{\mathcal{O}}\big(\sqrt{nd(K\sum_{\ell=1}^L \sigma_\ell^2 + L\sigma_{L+1}^2)}\big), \quad \texttt{HierTS-2}: \tilde{\mathcal{O}}\big(\sqrt{nd(K\sigma_1^2 + \sum_{\ell=1}^L \sigma_{\ell+1}^2)}\big).
$$

When $K \approx L$, the regret bounds of HierTS-1 and HierTS-2 are similar. However, when $K > L$, HierTS-2 outperforms HierTS-1. This is because HierTS-2 puts more uncertainty on a single $d$-dimensional latent parameter $\psi_{*,1}$, rather than $K$ individual $d$-dimensional action parameters $\theta_{*,i}$. More importantly, HierTS-1 implicitly assumes that action parameters $\theta_{*,i}$ are conditionally independent given $\psi_{*,L}$, which is not true. Consequently, HierTS-2 outperforms HierTS-1. Note that, under the linear diffusion model (15), dTS and HierTS-2 have roughly similar regret bounds. Specifically, their regret bounds dependency on $K$ is identical, where both methods involve multiplying $K$ by $\sigma_1^2$, and both enjoy improved performance compared to HierTS-1. That said, note that Theorem 4.1 and Proposition 4.2 provide an understanding of how dTS's regret scales under linear score functions $f_\ell$, and do not say that using dTS is better than using HierTS when the score functions $f_\ell$ are linear since the latter can be obtained by a proper marginalization of latent parameters (i.e., HierTS-2 instead of HierTS-1). While such a comparison is not the goal of this work, we still provide it for completeness next.

When the mixing matrices $\mathrm{W}_\ell$ are dense (i.e., assumption (A3) is not applicable), dTS and HierTS-2 have comparable regret bounds and computational efficiency. However, under the sparsity assumption (A3) and with mixing matrices that allow for conditional independence of $\psi_{*,1}$ coordinates given $\psi_{*,2}$, dTS enjoys a computational advantage over HierTS-2. This advantage explains why works focusing on multi-level hierarchies typically benchmark their algorithms against two-level structures akin to HierTS-1, rather than the more competitive HierTS-2. This is also consistent with prior works in Bayesian bandits using multi-level hierarchies, such as Tree-based priors [Hong et al., 2022a], which compared their method to HierTS-1. In line with this, we also compared dTS with HierTS-1 in our experiments. But this is only given for completeness as this is not the aim of Theorem 4.1 and Proposition 4.2. More importantly, HierTS is inapplicable in the general case in (1) with non-linear score functions since the latent parameters cannot be analytically marginalized.

# E    Broader impact

This work contributes to the development and analysis of practical algorithms for online learning to act under uncertainty. While our generic setting and algorithms have broad potential applications, the specific downstream social impacts are inherently dependent on the chosen application domain. Nevertheless, we acknowledge the crucial need to consider potential biases that may be present in pre-trained diffusion models, given that our method relies on them.

# F    Limitations

Our work investigated contextual bandits, laying the groundwork for future exploration into reinforcement learning. This exploration can be done from both practical (empirical) and theoretical angles. While our method, which approximates rewards using a Gaussian distribution, worked well for linear rewards and those following a generalized linear model, its effectiveness in real-world, complex scenarios needs further testing. Another interesting direction for future research is pre-training the diffusion model prior. Hsieh et al. [2023] proposed a method for this in multi-armed bandits, but its application to contextual bandits remains unexplored.

# G    Amount of computation required

Our experiments were conducted on internal machines with 30 CPUs and thus they required a moderate amount of computation. These experiments are also reproducible with minimal computational resources.

