# OpenReview forum: "Diffusion Models Meet Contextual Bandits with Large Action Spaces"
_NeurIPS.cc/2024/Conference — Submitted to NeurIPS 2024_

### Official Review · Reviewer_ByDg · 2024-07-02

**Soundness:** 3
**Presentation:** 4
**Contribution:** 3
**Rating:** 7
**Confidence:** 2

**Summary:**

The authors propose diffusion Thompson sampling, which uses a diffusion model to leverage reward under similar actions for more efficient exploration. The authors derive efficient posterior approximations under a diffusion model prior and prove a regret bound in linear instances. To efficiently compute and sample posterior distribution, the authors provide an approximation that relies on close-form solutions for case where both the score functions of the diffusion model and the likelihood are linear. For nonlinear diffusion model, the authors approximate posteriors by a Gaussian distribution.

**Strengths:**

The proof of Theorem 4.1 requires novel techniques such as recursive total variance decomposition and refined arguments such as quantifying not only the posterior information gain for the taken action but also for every learnt latent parameter.

The paper is well-written. The main contributions and key observations from the regret bound are nicely summarized. Experimental results for all four combinations of linear and nonlinear reward, linear and nonlinear diffusion model are provided. In experiments, the authors made a number of insightful observations, accompanied by ablation results.

**Weaknesses:**

The authors discussed how the number of layers L affect the regret bound. A higher L increases regret bound and a smaller L may fail to capture a more complex prior. It would improve the paper to provide a heuristics on choosing an appropriate L along with justifications for the heuristics.

**Questions:**

Please see weakness above.

**Limitations:**

The authors addressed limitations and societal impact at Appendix E and F.

---

> ### Author Rebuttal · Authors · 2024-08-04
>
> Thank you for your positive feedback and valuable time. Below, we provide our response to your question regarding the heuristic for choosing $L$.
>
> **Heuristic for Choosing $L$**
>
> Your intuition is correct: a higher $L$ increases the regret bound, while a smaller $L$ may result in a prior that fails to fully capture the true distribution of action parameters. Providing a theoretical heuristic for choosing $L$ is challenging, as it depends on various factors such as the complexity of the bandit problem (e.g., the dimension of the action parameters and the complexity of their true distribution, etc.). For example, in recommender systems where actions correspond to products, the choice of $L$ would differ between a diffusion model pre-trained on product image data and one trained on small $d$-dimensional product features.
>
> We can, however, tune $L$ empirically by finding the smallest $L$ that accurately captures the distribution of offline data. This tuning can be performed offline before integrating the diffusion model prior into dTS. In our additional simulations, we set $d=2$ and $K=100$. Unlike our main experiments, the true action parameters are sampled from the Swiss roll distribution (Figure 1 in the attached PDF in our global response), not a diffusion model. The diffusion model used by dTS is pre-trained on samples from this distribution.
>
> In Figure 2 (b) in the attached PDF in our global response, $L \approx 40$ leads to the best performance. Beyond $L \approx 40$, performance did not improve. Although our theory doesn't directly apply here since the diffusion model isn't linear, it can offer some intuition. Recall that dTS regret bound increases with $L$ assuming the true distribution is a (linear) diffusion model. When $L$ is small, the diffusion model doesn't fully capture the true distribution, violating this assumption. However, starting from $L \approx 40$, the true distribution is nearly captured, and increasing $L$ leads to higher regret, aligning with our theorem. This discussion isn't rigorous since our theorem's linear diffusion assumption isn't met, but it could explain decreased performance for $L$ higher than 40.
>
> We hope this discussion and our experiments address your question and increase your confidence in our work. Thank you again for your positive feedback.

---

> > ### Comment · Reviewer_ByDg · 2024-08-13
> >
> > I thank the authors for the response. The discussion on choosing L is a nice addition to the work. I will keep my score.

---

### Official Review · Reviewer_YAin · 2024-07-10

**Soundness:** 3
**Presentation:** 3
**Contribution:** 2
**Rating:** 6
**Confidence:** 4

**Summary:**

This work presents the use of Diffusion models as priors for Thompson sampling.

Namely, they propose to learn diffusion models (as replacement to other parametric priors) to accommodate more complex correlations between context, action and reward functions than with simple parametric form priors.

Given that Thompson sampling requires sampling from the posterior of the model, the authors derive a linear-Gaussian posterior approximation (under the proposed diffusion model prior).

The authors analyze the proposed algorithm for the linear-Gaussian reward case, which enables them to provide a Bayes regret bound.

Experimental results demonstrate some of the benefits of the proposed diffusion-based Thompson sampling: learning the correct latent-structure is beneficial, learning more parameters (as a function of $d$, $K$ and $L$) is a harder problem, hence incurs in higher regret.

**Strengths:**

- The use of diffusion models to learn complex priors for their use within MAB problems is of interest and significant.

- The authors provide a theoretical analysis of their proposed algorithm (only for the linear-Gaussian case), for which they:
    - use the recursive total covariance decomposition,
    - showcase the dependency over K ---induced by the hierarchical parameter learning--- and
    - demonstrate the dependency with $L$, inherent to having more parameters to learn.

- The theoretical analysis and the experiments showcase the benefits of learning the true hierarchical model (as specified by a diffusion model) in comparison to LinTS.

**Weaknesses:**

- The proposed diffusion-based algorithm does not learn the diffusion model as it sequentially interacts with the world
    - Instead, using the diffusion model as a complex prior requires offline learning, so that non-trivial prior distributions can be learned, before it can be used within Thompson sampling.
    - The cost of learning such a diffusion model is not acknowledged nor discussed.

- The proposed posterior approximation seems to be equivalent to the well known Laplace approximation, i.e., a linear-Gaussian approximation to a (non-linear and non-Gaussian) posterior. See questions below.

- The provided Bayesian regret is limited to the linear-Gaussian case, and in fact is acknowledged to be similar to "L + 1 sequential linear bandit instances stacked upon each other".

- The empirical evaluation is executed on synthetic experiments simulated from the assumed model prior, with $L$ latent parameters. Hence, the benefits of learning the true model are somehow expected.

**Questions:**

- The authors present this work as a new bandit algorithm that merges diffusion models with Thompson sampling.
    - However, one must "pre-train a diffusion model on offline estimates of the action parameters", and then use such diffusion model as a prior for Thompson sampling.
    - Hence, is the novelty of this work on proposing to use a flexible prior, that is learned offline, which is then applied to TS?
    - **Authors have clarified in their rebuttal how they pre-train the diffusion model offline, and update it online within TS**

- How does the proposed linear-Gaussian approximation to posteriors with non-Gaussian likelihoods in Equation (7) and non-linear functions in Section 3.2 relate to the Laplace approximation?
   - Such approximation is well known in the nonlinear state-space literature, can the authors please clarify the connection and significance?
   - **Clarified in the response**

- The authors claim to "demonstrate that with diffusion models parameterized by linear score functions and linear rewards, we can derive exact closed-form posteriors without approximations".
    - How is this result different from the well known result that linear combinations of Gaussian distributions are still Gaussian distributions?
    - **Addressed in rebuttal**

- Can the authors provide details on what is the offline procedure to learn the diffusion model prior in their synthetic experiments?
    - How much data did they use to learn these?
    - How sensitive is the procedure to poorly fitted diffusion priors?
    - **Addressed in rebuttal, please incorporate details and experiments to final manuscript**

- Even if the authors provide some results under the misspecified case in Section 5.2.3), can the authors provide more empirical results showcasing how the evaluated models perform when the true underlying model is different?
    - E.g., when there is a misspecification on the number of latent parameter layers $L$?
    - E.g., when the actual model does not have latent parameters, i.e., the assumptions made by LinTS?
       - This would help assess the impact of learning a complex diffusion model, when simpler parametric priors suffice.
    - **Addressed in rebuttal, please incorporate details and experiments to final manuscript**

- The authors refer to $f_l$ (the mapping of parameters at layer $l$ to the expected value of the parameters at layer $l-1$) as the "score function" . In statistics, the score function is often referred to as the derivative of the log-likelihood function with respect to the parameter. Are there any connections here that motivate the authors to use such naming? Clarifications on this matter would be appreciated.
    - **Clarified in their rebuttal**

**Limitations:**

The authors do present general limitations of their work, although the cost associated with learning a diffusion model prior is less clear.

---

> ### Author Rebuttal · Authors · 2024-08-06
>
> Thank you very much for your positive feedback and time. We provide point-by-point responses to your comments.
>
> **Offline Prior Pre-Training**
>
> Current experiments do not include diffusion model pre-training. Since the true distribution of action parameters is defined by a diffusion model, we directly use that model in dTS. However, including experiments with pre-training is important. We conducted these experiments in our global response above (see points __(1), (2), and (3)__), and here are additional clarifications specific to your questions.
>
> - **How much data did they use to learn these?** In our simulations, dTS outperforms LinTS by a factor of 1.5 using as few as 50 pre-training samples.
>
> - **How sensitive is the procedure to poorly fitted diffusion priors?** More pre-training samples lead to improved performance. But even with a poorly fitted diffusion model (e.g., pre-trained on only 50 samples, as shown in Figure 1(a) in the attached PDF), dTS outperforms LinTS because it still captures the Swiss roll structure better than a Gaussian distribution.
>
> - **Misspecification:**
>   - **Swiss Roll Simulations.** The true distribution of action parameters is not a diffusion model, so the diffusion model prior is misspecified, especially when learned with few pre-training samples. Despite this, dTS outperforms LinTS, which assumes a Gaussian prior. The effect of $L$ is studied in Figure 2(b) of the attached PDF, where $L \approx 40$ leads to the best performance.
>   - **MovieLens Simulations.** MovieLens problems are not sampled from a diffusion model (a user is sampled from the rating matrix, and the reward is the user's rating of a movie). Yet, dTS with a pre-trained diffusion model on embeddings from low-rank factorization of the rating matrix performs very well.
>   - **Existing Misspecification Experiments:** These experiments complement the misspecification experiments in the main text, which include likelihood and diffusion model parameter misspecification.
>
> __The Diffusion Model is Pre-trained Offline and Further Learned Online.__
>
> dTS uses a diffusion model prior that is pre-trained on offline data but also updates this model as it sequentially interacts with the environment. The process is as follows: we start with a pre-trained diffusion model as the prior for dTS. As dTS interacts with the environment, it updates this diffusion model. The "updated diffusion model" corresponds to the posterior derived in Section 3.2, with sequentially updated parameters. This is illustrated in Figures 3(a) and 3(b) in the attached PDF. The red samples in Figure 3(a) are from the pre-trained diffusion model (before dTS interacts with the environment), while the red samples in Figure 3(b) are from the diffusion model after 100 interactions. Clearly, the diffusion model is updated and becomes more concentrated.
>
> __Novelty__
>
> We use a pre-trained diffusion model as a strong and flexible prior for dTS. Diffusion model pre-training relies on offline data which is often widely available. This diffusion model is then sequentially refined through online interactions using our posterior approximation. This approximation allows fast sampling and updating of the posterior while performing very well empirically. dTS regret is bounded in a simple linear instance.
>
> __When the True Distribution of Action Parameters is Gaussian (Figure 3 (c) in the attached PDF)__
>
> We conducted an experiment where action parameters are drawn from $\mathcal{N}(0_d, I_d)$ with $d=2$ (Figure 3(a) in the attached PDF), and this prior distribution is provided to LinTS, meaning LinTS knows the true distribution. dTS, on the other hand, pre-trains a diffusion model using 1,000 samples from $\mathcal{N}(0_d, I_d)$. In this case, dTS performs comparably to LinTS (Figure 3(c) in the attached PDF). This is the best a TS variant can achieve in this scenario where LinTS has access to the true distribution.
>
> __Connection to Laplace Approximation.__
>
> The Laplace approximation approximates the posterior by a Gaussian distribution. In contrast, our approximate posterior remains a diffusion model with updated parameters, not a Gaussian. The only place we use a similar approximation is in line 140, where the likelihood is approximated by a Gaussian. Instead of approximating the entire posterior by a Gaussian, we only approximate the likelihood by a Gaussian and propagate this through the diffusion model, as explained in Section 3.2. We will clarify this connection in the revised manuscript.
>
> __Connection to Linearity in Gaussians.__
>
> As mentioned in line 128, (1) becomes a linear Gaussian system (LGS) [1] when the diffusion model is linear. Our derivations rely on the fact that a linear combination of Gaussians remains Gaussian. However, careful derivations were still necessary to obtain the correct posterior expressions, which is why we included them in the Appendix. We will change "we demonstrate that" in our statement to "we remark".
>
> __Score function__
>
> Great remark! We borrowed the term "score function" from the diffusion model literature, where it refers to the gradient of the log probability density of the data at a specific step $\ell$ w.r.t. the data itself. This differs from the standard definition in statistics, where the gradient is taken w.r.t. parameters. In diffusion models, the score function indicates the direction in which the noisy data should be adjusted to reduce noise. We used this term to denote $f_\ell$ as an abuse of terminology, but to avoid confusion, we will simply call it "function $f_\ell$".
>
> We're grateful for the detailed feedback. It allowed us to strengthen our work through additional experiments and discussions. If we've adequately addressed the reviewer's concerns, a re-evaluation of our work and a potential increase in score would be greatly appreciated. For any unresolved issues, we're happy to engage further. Thank you very much!
>
> [1] Pattern Recognition and Machine Learning(2006), Christopher M. Bishop.

---

> > ### Comment · Reviewer_YAin · 2024-08-09
> > **Thank you for your clarifying response!**
> >
> > I appreciate the authors' clear and informative response to my questions:
> >
> > - *Diffusion Model training*: The provided response now clarifies that the Diffusion model is pre-trained offline (please incorporate all the rebuttal details and additional experiments) and then further learned online.
> >
> > - *Connection of derivation with state-of-the-art*: I appreciate the authors careful delineation of what is new in their work and where are the connections with other known approaches.
> >
> > - *Score function*: thanks for the clarification!
> >
> > Given the informative response to my question and other reviewers (please incorporate new details, discussions and experiments into the updated manuscript), I am increasing my score.

---

### Official Review · Reviewer_wiX3 · 2024-07-13

**Soundness:** 4
**Presentation:** 4
**Contribution:** 4
**Rating:** 8
**Confidence:** 5

**Summary:**

The work provides a great example of diffusion modeling on bandit action parameter for better exploration.

**Strengths:**

The work provides a comprehensive description on how to employ diffusion modeling on bandit parameters for contextual bandit problems.

The discussion on linear and non-linear diffusion model is clean and precise for readers with background in Thompson Sampling

The analysis part also provide comprehensive discussion on how the regret of the  proposed diffusion Thompson Sampling scales with main dimension of contextual bandit problems.

**Weaknesses:**

I am satisfied with current version of the paper.

**Questions:**

(1) Could you discuss how to validate assumption (A1)(A2)(A3) in practice and also contrast the assumptions in the literature?

(2) Could you discuss on some strategy to prove frequentist regret for diffusion Thompson sample?

**Limitations:**

Yes, the author clearly states the assumptions to address the limitation of the theoretical analysis.

---

> ### Author Rebuttal · Authors · 2024-08-03
>
> We sincerely appreciate your positive feedback and recognition of our work. Below, we provide our responses to your comments.
>
> **Assumptions**
>
> - **(A1)** is common in the literature and can be easily satisfied in practice by normalizing contexts. For example, in a recommender system, normalizing user features (which correspond to contexts) before inputting them into TS would suffice. This assumption can be relaxed to any context $X_t$ with bounded norms $||X_t||$.
>
> - **(A2)** has two parts. The first part assumes that the covariance matrices can be expressed as $\Sigma_\ell = \sigma_\ell^2 I_d$, which can be relaxed to any positive definite covariances. This is similar to the assumption in the Bayesian regret analysis of LinTS where the prior is Gaussian with a positive-definite covariance. The second part, $\lambda_1(W_\ell^\top W_\ell) = 1$, is specific to the parametrization of the diffusion model prior but can be relaxed to any arbitrary matrices $W_\ell$. It was made to ease the exposition only.
>
> - **(A3)** aims to tighten the regret bound (improved dependence on the dimension $d$). It assumes that the matrices $W_\ell$ are low-rank. A similar assumptions is made in the low-rank bandit literature (see discussion starting in line 589 in Appendix A).
>
> **Frequentist Regret**
>
> A frequentist regret bound is indeed possible, requiring only a new proof for Lemma D.1, potentially using martingale bounds for tail events and anti-concentration bounds for posterior sampling akin to [1,2]. The remainder of our regret proof, which relies on worst-case arguments, would remain unchanged. While we are confident in the feasibility of this extension, it necessitates careful consideration to rigorously address any technical challenges that could arise. This is very interesting for future research, and we will add it to our discussion.
>
> Thank you for your valuable time, and please let us know if you have any further questions.
>
> [1] Agrawal, Shipra, and Navin Goyal. "Thompson sampling for contextual bandits with linear payoffs." International conference on machine learning. PMLR, 2013.
>
> [2] Abeille, Marc, and Alessandro Lazaric. "Linear Thompson sampling revisited." Artificial Intelligence and Statistics. PMLR, 2017.

---

### Official Review · Reviewer_89g4 · 2024-07-21

**Soundness:** 3
**Presentation:** 3
**Contribution:** 2
**Rating:** 5
**Confidence:** 4

**Summary:**

The paper considers the problem of contextual bandits in large action spaces.  In this problem, the reward of an arm is a function of the context and an unknown, arm specific parameter vector. To efficiently learn good policies in such large action spaces, the paper places a structured-prior distribution on the unknown arm parameters that can effectively capture the correlations between the arms. The specific form of the prior distribution considered in the paper resembles a diffusion model. The main contribution of the paper is to provide a computationally efficient heuristic for performing Thompson sampling with this prior. Experiments on synthetic data show that the proposed technique is much better at learning optimal policies than other popular baselines such as LinTS, LinUCB.

**Strengths:**

The problem of handling large action spaces in contextual bandits seems interesting. The empirical evaluation shows promise in the proposed approach

**Weaknesses:**

- **Related Work:** There are several ways in which large action spaces are typically handled in contextual bandits. One popular approach that is used in practice is to associate a feature vector to each arm (this feature vector is known to the learner ahead of time), and the reward of pulling certain arm for a context is a function of both the context and arm features. In the absence of arm features, the other approach is to impose some structure on the unknown arm parameter vectors. There are several works which do this, and the current paper falls in this line of research. Some of these works assume the arms can be clustered into a small number of groups or can be embedded in a low-dimensional latent space and learn the low-dimensional features during the course of the online learning (https://arxiv.org/pdf/2010.12363, https://arxiv.org/abs/2209.03997, https://arxiv.org/pdf/1810.09401, https://proceedings.neurips.cc/paper_files/paper/2023/file/f334c3375bd3744e98a0ca8eaa2403b0-Paper-Conference.pdf). The diffusion prior used in the current work resembles the low-dimensional embedding assumption. In particular, it is assumed there is a latent vector (psi_1) from which all the arm parameter vectors are generated; this is a form of rank-1 assumption on the arm features. Unfortunately, none of these works were brought up in the paper. It would be great if the authors perform a thorough literature review and better position their work.

- **Linear Setting**: A lot of emphasis has been placed on the linear model in the paper. I understand it is used to derive the heuristic for the non-linear setting. Beyond that, I do not find the regret bounds derived in section 4 to be interesting. In the linear setting, there isn't a need to work with the complex hierarchical diffusion prior. It looks like one could totally remove the latent variables psi_{*, L}, .... psi_{*, 2} and simply place a Gaussian prior on psi_{*,1} and get an equally powerful model. This would also improve the regret bounds, by removing the L factor in the regret. Given this, I'm not sure about the utility of section 4.
  - Section 4.1 compares the regret bounds obtained in this work with other baselines. But this comparison is only meaningful under the assumption that the diffusion prior is properly specified. This raises the following question: why is this a reasonable prior to use in practice? How do various techniques compare if this prior is misspecified? (There are some experiments section 5.2 on prior misspecification, but the misspecifications considered there seem to be very minor)

- **Quality of Heuristics:** How good is the heuristic used for non-linear diffusion model? There is no discussion on this in the paper (In my opinion, this needs to be thoroughly discussed in the paper, as it is the primary novelty of the work). Some empirical evaluation comparing it with other standard estimation techniques (such as variational techniques, and other posterior estimation techniques) would have been helpful in understanding his question.

**Questions:**

- **Choice of prior**: What is the reason behind choosing this particular choice of prior?
- **Offline samples for estimating prior**: Line 33 says that offline estimates of parameters are leveraged to build a diffusion model. But this point was never brought up in the rest of the paper. What is the problem setting considered in this work? If the prior is estimated from offline data, then more information needs to be provided on how this is done and how many samples are needed to get a good estimate.
- **Non-linear diffusion model:** Can something be said about the quality of the approximation in the asymptotic setting where the samples go to infinity?

**Limitations:**

See my comments above

---

> ### Author Rebuttal · Authors · 2024-08-05
>
> Thank you very much for your valuable feedback and time. We provide point-by-point responses to your comments.
>
> __Offline Samples for Prior Pre-Training.__
>
> We address this question in our global response above (please see points __(1)__, __(2)__ and __(3)__).
>
> __Related Work.__
>
> Thanks for providing these references. Similar papers were discussed in Appendix A, starting from line 589 and we will incorporate these references into that discussion, and mention them in the main text to reflect their relevance to our work.
>
> __Linear Setting.__
>
> We agree that in the fully linear-Gaussian setting (linear diffusion and linear rewards), placing a properly chosen Gaussian prior on $\psi_1$ suffices. However, this comparison is not the purpose of Section 4. Instead, Section 4 aims to evaluate the performance of dTS in a simplified setting, which allows for theoretical analysis due to exact posteriors. It highlights the dependence of dTS's regret bound on problem parameters and does not claim that dTS should be preferred over placing a Gaussian prior on $\psi_1$ in this simplified setting. Rather, it complements the paper by analyzing a simplified instance of dTS, with the paper's overall contributions extending way beyond this setting and analysis.
>
> __Choice of Prior.__
>
> Due to their recent success in capturing complex structures (e.g., image generation) and advances in decision-making applications, we chose diffusion models as priors for TS. Their special form allows us to derive efficient approximations that are nearly in closed form (for linear rewards, they are in closed form, and for non-linear rewards, they are nearly in closed form with only the MLE being approximated through numerical optimization). The key idea is that starting from a sufficiently good prior accelerates TS online learning. If a good enough prior in a specific application is something other than diffusion models, it can certainly be used, but different posterior approximations will be needed. Currently, diffusion models achieve state-of-the-art performance in capturing complex distributions and allow efficient posterior approximations, which is why we selected them.
>
> __Asymptotic Behavior of Our Approximation.__
>
> Our approximation retains a key attribute of exact posteriors: they match the prior when there is no data, and the effect of the prior diminishes as data accumulates. Therefore, in the asymptotic setting, the posterior is Gaussian with mean equal to the MLE and covariance 0. An exact posterior would have similar asymptotic behavior (Bernstein-von Mises Theorem assuming some regularity conditions are satisfied).
>
> __Quality of Our Posterior Approximation  (Figures 3 (a, b) in the attached PDF in our global response).__
>
> To assess the quality of our posterior approximation, we consider the scenario where the true distribution of action parameters is $\mathcal{N}(0_d, I_d)$ with $d=2$ and rewards are linear. We pre-train a diffusion model using samples drawn from $\mathcal{N}(0_d, I_d)$. We then consider two priors: the true prior $\mathcal{N}(0_d, I_d)$ and the pre-trained diffusion model prior. This yields two posteriors:
>
> - **$P_1$**: Uses $\mathcal{N}(0_d, I_d)$ as the prior. $P_1$ is an exact posterior since the prior is Gaussian and rewards are linear-Gaussian.
> - **$P_2$**: Uses the pre-trained diffusion model as the prior. $P_2$ is our approximate posterior.
>
> The learned diffusion model prior matches the true Gaussian prior (as seen in Figure 3 (a)). Thus, if our approximation is accurate, their posteriors $P_1$ and $P_2$ should also be similar. This is observed in Figure 3 (b) where the approximate posterior $P_2$ nearly matches the exact posterior $P_1$.
>
> Other approximation methods can be used, but they can be costly. We need _fast updates of the posterior_ and _fast sampling from the posterior_, both of which our approximation achieves. For linear rewards, the formulation is in closed form, requiring no optimization, and for non-linear rewards, only the MLE requires numerical optimization. These two requirements may not be met by other methods. For example, optimizing a variational bound using the re-parameterization trick and Monte Carlo estimation would introduce a complex optimization problem into a bandit algorithm that needs to be updated in each interaction round.
>
> We're grateful for the feedback, which has allowed us to strengthen our work through additional experiments and discussions. If we've adequately addressed the reviewer's concerns, a re-evaluation of our work and a potential increase in score would be greatly appreciated. For any unresolved issues, we're happy to engage further. Thank you very much!

---

### Author Rebuttal · Authors · 2024-08-05

We are very grateful to the reviewers and AC for their valuable time. This global response includes additional experiments and discussions on the impact of pre-training on dTS performance (Reviewers 89g4 and YAin). We have included a PDF with figures related to these experiments.

The attached PDF also contains experiments assessing the impact of diffusion depth $L$ on dTS performance (Reviewer ByDg), experiments where action parameters are sampled from a simple Gaussian distribution (Reviewer YAin), and experiments assessing the quality of our posterior approximation (Reviewer 89g4). The explanations and discussions for these three experiments are provided in our individual responses to the respective reviewers.

**(1) Offline Pre-training Procedure**

We used JAX for diffusion model pre-training, summarized as follows:

- **Parameterization:** Functions $f_\ell$ are parameterized with a fully connected 2-layer neural network (NN) with ReLU activation. The step $\ell$ is provided as input to capture the current sampling stage. Covariances are fixed (not learned) as $\Sigma_\ell = \sigma_\ell^2 I_d$, with $\sigma_\ell$ increasing with $\ell$.

- **Loss:** Offline data samples are progressively noised over steps $\ell \in [L]$, creating increasingly noisy versions of the data following a predefined noise schedule [1]. The NN is trained to reverse this noise (i.e., denoise) by predicting the noise added at each step. The loss function measures the $L_2$ norm difference between the predicted and actual noise at each step, as explained in [1].

- **Optimization:** Adam optimizer with a $10^{-3}$ learning rate was used. The NN was trained for 20,000 epochs with a batch size of min(2048, pre-training sample size). We used CPUs for pre-training, which was efficient enough to conduct multiple ablation studies.

- **After pre-training:** The pre-trained diffusion model is used as a prior for dTS and compared to LinTS as the reference baseline. In our ablation study, we plot the cumulative regret of LinTS in the last round divided by that of dTS. A ratio greater than 1 indicates that dTS outperforms LinTS, with higher values representing a larger performance gap.

__(2) Impact of Pre-Training Sample Size (Figure 1 and Figure 2 (a) in attached PDF)__

Unlike our main experiments, the true action parameters are sampled from the Swiss roll distribution (Figure 1 in the attached PDF in our global response), not a diffusion model. The diffusion model used by dTS is pre-trained on samples from this distribution.

Figure 2 (a) shows that higher sample sizes increase the performance gap between dTS and LinTS. More samples improve the diffusion prior's estimation (see Figure 1), leading to better dTS performance. Remarkably, we observed comparable performance with as few as 10 samples, and dTS outperforms LinTS by a factor of 1.5 with just 50 samples. More samples would be needed for more challenging problems, but LinTS would also struggle in such cases. Thus, we expect these gains to be even more pronounced in more complex settings.

__(3) MovieLens Experiments (Figure 2 (c) in attached PDF)__

We also evaluate dTS using the standard MovieLens setting. In this semi-synthetic experiment, a user is sampled from the rating matrix in each interaction round, and the reward is the rating that user gives to a movie (see Section 5 in [2] for details about this setting). Here, the true distribution of action parameters is unknown and not a diffusion model. The diffusion model is pre-trained on offline estimates of action parameters obtained through low-rank factorization of the rating matrix. Figure 2(c) shows that dTS outperforms LinTS in this setting.

__References__

[1] Ho, J., Jain, A., & Abbeel, P. Denoising diffusion probabilistic models. NeurIPS 2020.

[2] Clavier, P., Huix, T., & Durmus, A. VITS: Variational Inference Thomson Sampling for contextual bandits. ICML 2024.

---

### Decision · Program_Chairs · 2024-09-25

**Decision:**

Reject

**Comment:**

This paper tackles the problem of contextual bandits with large action spaces by taking a diffusion model approach for modelling the prior on the unknown arm parameters. The goal is to make exploration more efficient and reduce the regret dependence on the number of arms while offering a computationally attractive algorithm.

After the rebuttal and discussion period, all reviewers are happy with the empirical evaluation part of the paper. However, reviewers have expressed concerns regarding how interesting it is to study the linear setting with Gaussian priors and how novel and different the resulting algorithm is in this setting compared to an algorithm which uses only a single layer. While the authors try to address precisely these points in the rebuttal, after a discussion period there are reviewers who still think that the results in the linear-Gaussian setting are of questionable impact. After reading the paper myself I am not completely convinced about the novelty of the proposed approach for the linear setting. In particular I find that the proof techniques are similar to Aouali et al. [2023b] and the theoretical contributions seem somewhat incremental. Further, from the regret bounds in Theorem 4.1 and Proposition 4.2 it is unclear how the approach actually solves the problem of large action spaces as the regret still scales with $\sqrt{K}$. While the authors try to explain this dependence in lines 215-219 I find this explanation unsatisfactory due to the following. A simple Taylor expansion around 0 for $\log(1+x)$ shows that the first term of the regret bound in Proposition 4.2 is about $\sqrt{n^2 K \sigma_1^2 / d}$. Now if $\sigma_1^2 = O(d/n)$ which is considerably small (notice it tends to 0 as the horizon goes to infinity), the regret bound still evaluates to $\sqrt{n K}$.

Given that the theory sections of the paper are a significant part of the contributions and that there still remain concerns regarding the novelty and impact of the theory I lean towards rejecting the current work.